# Photolytic modification of seasonal nitrate isotope cycles in East Antarctica

Pete D. Akers[1,2], Joël Savarino[2], Nicolas Caillon[2], Olivier Magand[2], Emmanuel Le Meur[2]

[1]Department of Geography, Trinity College Dublin, Dublin, Ireland.
[2]Université Grenoble Alpes, CNRS, IRD, Grenoble INP, IGE, Grenoble, France.

Correspondence to: Pete D. Akers (pete.akers@tcd.ie)

**Abstract.** Nitrate in Antarctic snow has seasonal cycles in nitrogen and oxygen isotopic ratios that reflect its sources and atmospheric formation processes, and as a result, nitrate archived in Antarctic ice should have great potential to record atmospheric chemistry changes over thousands of years. However, sunlight that strikes the snow surface results in photolytic

nitrate loss and isotopic fractionation that can completely obscure the nitrate's original isotopic values. To gain insight into how photolysis overwrites the seasonal atmospheric cycles, we collected 244 snow samples along an 850 km transect of East Antarctica during the 2013–2014 CHICTABA traverse. The CHICTABA route's limited elevation change, consistent distance between the coast and the high interior plateau, and intermediate accumulation rates offered a gentle environmental gradient ideal for studying the competing pre- and post-depositional influences on archived nitrate isotopes. We find that

nitrate isotopes in snow along the transect are indeed notably modified by photolysis after deposition, and drier sites have more intense photolytic impacts. Still, an imprint of the original seasonal cycles of atmospheric nitrate isotopes is present in the top 1–2 m of the snowpack and likely preserved through archiving in glacial ice at these sites. Despite this preservation, reconstructing past atmospheric values from archived nitrate in similar transitional regions will remain a difficult challenge without having an independent proxy for photolytic loss to correct for post-depositional isotopic changes. Nevertheless,

nitrate isotopes should function as a proxy for snow accumulation rate in such regions if multiple years of deposition are aggregated to remove the seasonal cycles, and this application can prove highly valuable in its own right.

## 1. Introduction

Nitrate ($NO_3^-$) is one of the most prevalent ions in Antarctic snow and ice, arriving as an end product of the atmospheric oxidation of nitrogen oxides ($NO_x = NO + NO_2$) in wet or dry deposition of nitric acid ($HNO_3$) or particulate nitrate (p-$NO_3^-$)

(Neubauer and Heumann, 1988; Wolff, 1995; Röthlisberger et al., 2000; Savarino et al., 2007; Frey et al., 2009; Shi et al., 2018b). Because the isotopic ratios of nitrogen and oxygen in atmospheric $NO_3^-$ reflect differences in the original sourcing of the $NO_3^-$ and its atmospheric reaction history, a long-term $NO_3^-$ archive could reveal how the atmosphere's oxidative capacity and chemical reaction pathways have changed over time (Legrand et al., 1999; Michalski et al., 2005; Wolff et al., 2007; Alexander et al., 2009; Kamezaki et al., 2019). Despite its paleoenvironmental potential, $NO_3^-$ has been difficult to

interpret in ice cores because post-depositional processes in the uppermost snowpack often result in substantial mass loss and isotopic changes (Wolff et al., 2002; Grannas et al., 2007; Frey et al., 2009; Erbland et al., 2013; Meusinger et al., 2014; Traversi et al., 2014; Geng et al., 2015). Before the paleoenvironmental potential of $NO_3^-$ can be fully realized, we require an improved understanding on how the isotopic values in $NO_3^-$ are altered during the archiving process in the snowpack from the atmospheric source into eventual glacial ice.

Atmospheric $NO_3^-$ sampled 1–10 m above the snow surface in Antarctica has clear annual cycles in concentration and isotopic values related to seasonal changes in $NO_3^-$ source and formation reaction pathways (Wagenbach et al., 1998; Savarino et al., 2007, 2016; Frey et al., 2009; Erbland et al., 2013; Ishino et al., 2017; Winton et al., 2020). Through wet or dry $NO_3^-$ deposition, these annual cycles are transferred with the $NO_3^-$ to the snow surface. After deposition, $NO_3^-$ photolysis, $HNO_3$ volatilization, and physical snow mixing can alter and obscure these cycles, but post-depositional $NO_3^-$

processes are largely restricted to a shallow (i.e., 0.1–1.0 m) surface layer of the snowpack where light can penetrate, interstitial air can exchange with the atmosphere, and snow can be eroded and mixed by wind (e.g., Grannas et al., 2007; Wolff et al., 2002; Röthlisberger et al., 2002; Frezzotti et al., 2002; Libois et al., 2014; Scarchilli et al., 2010; Picard et al., 2019). After $NO_3^-$ in a snow layer is buried beneath this "active zone" by additional snow accumulation, it is believed to be generally nonreactive and stable.

As a result, the magnitude of post-depositional isotopic changes relative to the initial depositional values is heavily controlled by the speed at which $NO_3^-$ is buried, i.e., the local surface mass balance (SMB, equivalent here to "net accumulation rate"). At very high SMB sites near the Antarctic coast, $NO_3^-$ is rapidly buried, and the original chemical nature of the atmospheric $NO_3^-$ is largely preserved through the burial process. At very low SMB sites, in contrast, it may take several years for $NO_3^-$ to be buried below the zone of active post-depositional processes, and $NO_3^-$ observed in ice cores

and snow pits at dry interior Antarctic stations has such substantial isotopic changes and extreme mass loss that the original depositional values of $NO_3^-$ are completely obscured (Freyer et al., 1996; Frey et al., 2009; Erbland et al., 2013; Shi et al., 2015). Most of Antarctica, however, falls between these two SMB extremes (Agosta et al., 2019), and archived $NO_3^-$ concentration and isotopic profiles throughout Antarctica likely exhibit a gradient between full preservation of the atmospheric $NO_3^-$ characteristics and the complete post-depositional loss of these characteristics. Snow and ice from

intermediate SMB sites can thus offer valuable insight into exactly how post-depositional processes interact with and change the initial isotopic chemistry of $NO_3^-$ that is deposited in Antarctica.

    We present here $NO_3^-$ data of snow samples taken during the CHICTABA ("Chemical-physical analyses of snow and firn for determining accumulation in Terre Adélie and Aurora Basin North") traverse across a lower elevation region of the East Antarctic Plateau in austral summer 2013–2014. The $NO_3^-$ data include $NO_3^-$ mass fractions ($\omega(NO_3^-)$), isotopic ratios

($\delta^{15}N_{NO3}$ and $\delta^{18}O_{NO3}$, where $\delta = \frac{R_{sample}}{R_{reference}} - 1$, with $R$ denoting the $^{15}N/^{14}N$ or $^{18}O/^{16}O$ isotopic ratios of $NO_3^-$, reported relative to the standards $N_2$-Air (Mariotti, 1983) and Vienna Standard Mean Ocean Water (VSMOW) (Baertschi, 1976), respectively), and the oxygen isotope anomaly ($\Delta^{17}O_{NO3}$, where $\Delta^{17}O_{NO3} = \delta^{17}O_{NO3} - 0.52 \times \delta^{18}O_{NO3}$) (Thiemens and Heidenreich, 1983). The sites sampled along this traverse have climatology and SMB intermediate to the coast and interior plateau, and thus the $NO_3^-$ offers an important link between existing studies focused on those two environments. With our

new data, we confirm the partial preservation of seasonal isotopic cycles, quantify isotopic fractionation due to post-depositional effects, and consider how these dual effects interact to produce the $NO_3^-$ values that will be archived into deeper ice.

## 2. Processes affecting $NO_3^-$ isotopic variability in Antarctica

### 2.1. Annual cycles in atmospheric $NO_3^-$ chemistry and sourcing

Seasonal changes of near surface atmospheric $NO_3^-$ concentration and isotopic ratios (Figure 1) are well-documented at multiple sites across East Antarctica (Wagenbach et al., 1998; Savarino et al., 2007; Frey et al., 2009; Erbland et al., 2013; Ishino et al., 2017; Xu et al., 2019; Winton et al., 2020; Shi et al., 2022a). Atmospheric $NO_3^-$ concentrations peak in late spring and early summer (Nov–Jan) and are 5–10 times lower in autumn and winter (Mar–Jul). Values of $\delta^{18}O_{NO3}$ and $\Delta^{17}O_{NO3}$ both peak in late winter (Jul–Sep) and are lowest in summer (Dec–Feb), resulting in a seasonal cycle that is offset

four months earlier from the $NO_3^-$ concentration cycle. The $\delta^{15}N_{NO3}$ values also vary seasonally, but with a less clear cycle. While the highest $\delta^{15}N_{NO3}$ values coincide with the late winter peak in $\delta^{18}O_{NO3}$ and $\Delta^{17}O_{NO3}$ values, the lowest $\delta^{15}N_{NO3}$ values occur in spring (Oct–Nov), 1–2 months before the minima in $\delta^{18}O_{NO3}$ and $\Delta^{17}O_{NO3}$. Additionally, a minor secondary peak in atmospheric $\delta^{15}N_{NO3}$ has also been observed at Dome C in January (Figure 1b) (Frey et al., 2009; Erbland et al., 2013; Winton et al., 2020).

These annual cycles have been attributed to changes in $NO_3^-$ sourcing and reaction pathways related to the distinctly different extreme environments of polar summer and winter. During daytime, photolysis can be a significant local source of $NO_3^-$ when ultraviolet solar radiation converts $NO_3^-$ in the snowpack into $NO_x$ gases that then ventilate upward into the atmosphere and oxidize back into $HNO_3$ (Frey et al., 2009; Erbland et al., 2015; Winton et al., 2020). In polar winter, however, the limited or complete lack of sunlight largely prevents photolysis from occurring, and atmospheric $NO_3^-$ over

Antarctica in winter is thought to be largely supplied through long-distance transport from lower latitudes (Savarino et al., 2007; Lee et al., 2014; Shi et al., 2018b; Walters et al., 2019). Substantial influx of this low latitude $NO_3^-$ is limited by the intense Antarctic polar vortex, and $NO_3^-$ concentrations in winter are very low as a result. During the coldest conditions in late winter and early spring, stratospheric denitrification through polar stratospheric cloud sedimentation supplies a small amount of $NO_3^-$ with relatively high $\delta^{15}N_{NO3}$, $\delta^{18}O_{NO3}$, and $\Delta^{17}O_{NO3}$ values to the troposphere above Antarctica (Fahey et al.,

1990; Van Allen et al., 1995; Santee et al., 2004; Savarino et al., 2007; Ishino et al., 2017; Shi et al., 2022a). This stratospheric supply produces a small observed increase in $NO_3^-$ concentration and contributes to the annual peaks in isotopic values (Figure 1). Additionally, because ozone ($O_3$) transfers its anomalously high $\Delta^{17}O$ value to $NO_3^-$ when it is involved in $NO_x$ cycling, the higher $\Delta^{17}O_{NO3}$ values observed in Antarctic winter are attributed to this $NO_3^-$ being sourced from lower latitudes and the stratosphere where $O_3$ oxidation is more important (Alexander et al., 2009; Savarino et al.,

2016; Ishino et al., 2017).

With the return of intense sunlight in spring, photolysis will convert much of the $NO_3^-$ that has accumulated in the near surface snowpack through winter into $NO_x$ which is rapidly re-oxidized into $HNO_3$ upon reaching the atmosphere (Wolff et al., 2002; Davis et al., 2004, 2008; Grannas et al., 2007; Jacobi and Hilker, 2007; Erbland et al., 2015; Winton et al., 2020; Barbero et al., 2021). This new source of "recycled" $NO_3^-$ produces a rapid rise in atmospheric $NO_3^-$ concentration in

November, with some $NO_3^-$ possibly supplied by additional recycled $HNO_3$ transported from upwind regions of Antarctica (Savarino et al., 2007; Shi et al., 2018a). The recycled $NO_3^-$ has isotopic values lower than the mean atmospheric $NO_3^-$ values due to strongly negative isotopic fractionation factors during $NO_3^-$ photolysis (Frey et al., 2009; Erbland et al., 2013; Berhanu et al., 2014, 2015; Shi et al., 2015) and incorporation of oxygen atoms from local water sources (snow and water vapor $\delta^{18}O = -20--80$ ‰, $\Delta^{17}O \approx 0$ ‰) during re-oxidation (McCabe et al., 2005; Erbland et al., 2013; Winton et al., 2020).

Sunlight also triggers additional oxidation pathways for $NO_3^-$ formation through $HO_x$, $RO_x$, and $H_2O_2$ that lack the anomalous $\Delta^{17}O$ value of $O_3$ (i.e., their $NO_3^-$ product has $\Delta^{17}O = 0$), and $\Delta^{17}O_{NO3}$ values are expected to decline in summer as these pathways compete with the $O_3$ pathway (Alexander et al., 2009; Savarino et al., 2016; Ishino et al., 2017). Several unknowns still exist based on disagreements between field observations and model predictions for isotopic values and photolytic constants, and the atmospheric $NO_3^-$ budget for Antarctica remains an active field of research (e.g., Savarino et

al., 2016; Walters et al., 2019; Barbero et al., 2021).

**2.2. Snow skin layer $NO_3^-$ chemistry**

The seasonal variability of $NO_3^-$ in the snowpack's "skin layer" (i.e., the uppermost 2–6 mm layer of loose snow grains) generally follows that of the local atmospheric $NO_3^-$ (Figure 1e-h). This similarity is because skin layer $NO_3^-$ is in a close exchange with atmospheric $NO_3^-$, being sourced from recently deposited atmospheric $NO_3^-$ and also supplying $NO_3^-$ to the

atmosphere through photolysis during sunlit times. Spatially across Antarctica, skin layer $\omega(NO_3^-)$ is generally higher at drier and more inland regions (Frey et al., 2009; Erbland et al., 2013; Shi et al., 2015, 2018b) despite atmospheric $NO_3^-$ concentrations showing far less spatial variability (Savarino et al., 2007; Frey et al., 2009; Shi et al., 2022a). The higher $\omega(NO_3^-)$ observed in the skin layer at drier sites is attributed to increased local $NO_3^-$ deposition from photolytic recycling as well as the fact that drier sites will dilute the $NO_3^-$ less when $NO_3^-$ deposition rates are similar across Antarctica (Erbland et

al., 2013; Shi et al., 2018b; Winton et al., 2020).

Some differences between atmospheric and skin layer values do exist, however. Notably, $\delta^{15}N_{NO3}$ values in the skin layer are 5–15 ‰ higher than the atmosphere, possibly due to isotopic fractionation as atmospheric $HNO_3$ gas adsorbs onto the snow surface (Erbland et al., 2013; Winton et al., 2020). Additionally, the $NO_3^-$ oxygen isotopes in the skin layer are consistently higher than those observed in atmospheric $NO_3^-$ (Erbland et al., 2013; Winton et al., 2020), and this unexpected discrepancy

is currently unexplained and puzzling. This difference is greatest in the early winter, when $\delta^{18}O_{NO3}$ and $\Delta^{17}O_{NO3}$ values can be up to 20 ‰ and 10 ‰ higher, respectively, in the skin layer than the atmosphere. Full annual skin layer observations of $\omega(NO_3^-)$ and $NO_3^-$ isotopes have until recently been only available from Dome C (Figure 1e–h) (Frey et al., 2009; Erbland et al., 2013; Winton et al., 2020), but a recent record from Zhongshan station suggests that oxygen isotopic values at coastal sites may match more closely between the atmosphere and snow surface (Shi et al., 2022a). Additional data from Zhongshan

station and other sites will allow us to better judge the representativeness of the Dome C data with regards to the broader Antarctic environment.

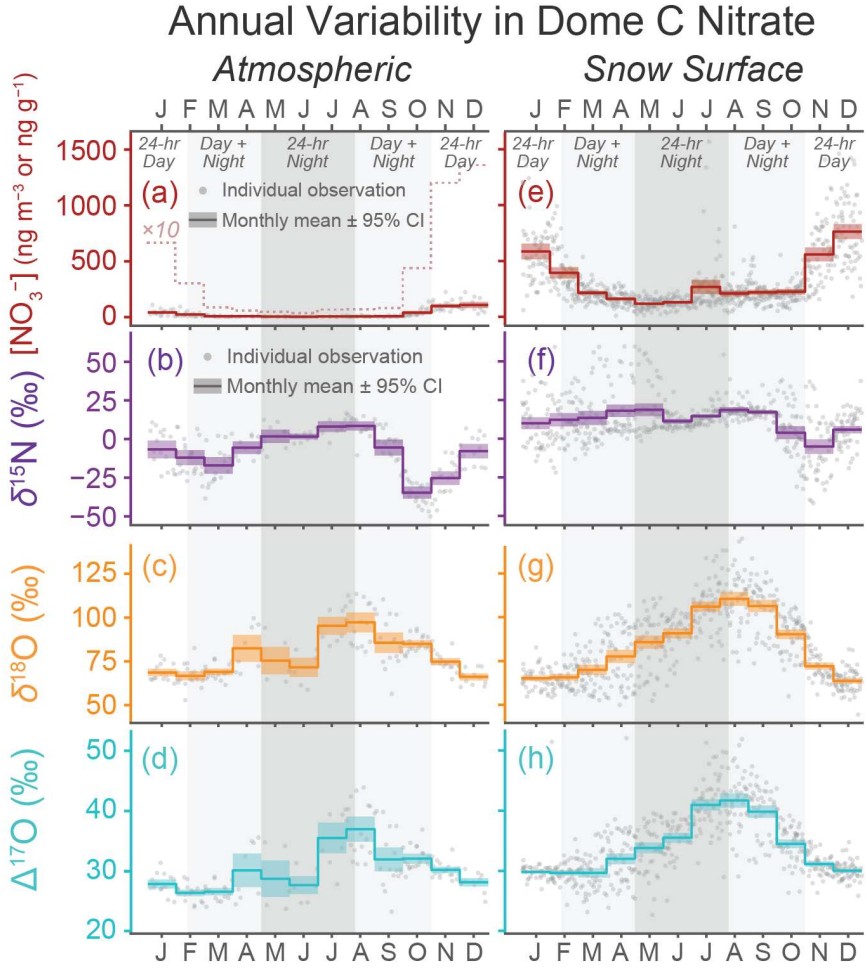

**Figure 1. Annual patterns of $NO_3^-$ variability in the atmospheric and snow surface at Concordia station, Dome C, Antarctica.** Data
shown covers previously reported samples taken in 2009–2014 (Erbland et al., 2013; Winton et al., 2020). Atmospheric $NO_3^-$ **(a–d)** was collected over week-long periods with a high-volume air filter located 5 m above the snow surface, and snow surface samples **(e–h)** were taken every 1–7 days from the 2–6 mm thick skin layer in the clean sector outside Concordia station. Individual points represent individual samples, and the thick colored lines represent the monthly mean values with the 95 % confidence interval of the mean shown as colored shading. Note that the units for $NO_3^-$ concentration is ng m$^{-3}$ for atmospheric $NO_3^-$ **(a)** and ng g$^{-1}$ for
the snow surface $NO_3^-$ **(e)**. A dashed line representing the atmospheric $NO_3^-$ concentration multiplied by 10 is included in **(a)** for better observation of the annual variation pattern.

**2.3 Post-depositional processes affecting NO$_3^-$**

During burial, several post-depositional processes can alter the values of skin layer NO$_3^-$. Past studies of buried NO$_3^-$ on the interior East Antarctic Plateau have highlighted photolysis as the primary post-depositional process that affects NO$_3^-$ in East Antarctic snow, resulting in substantial NO$_3^-$ mass loss that can reach > 90 % reduction at dome summits. The NO$_3^-$ remaining in snow after mass loss shows marked increases in $\delta^{15}N_{NO3}$ values due to a negative photolytic isotopic fractionation factor for nitrogen. Fractionation factors for $\delta^{18}O_{NO3}$ and $\Delta^{17}O_{NO3}$ are theoretically predicted to be negative, and therefore oxygen isotopic values of remaining NO$_3^-$ should increase like $\delta^{15}N_{NO3}$ after photolysis (Frey et al., 2009). However, NO$_3^-$ at sites with clear photolytic mass loss typically has $\delta^{18}O_{NO3}$ and $\Delta^{17}O_{NO3}$ values lower than atmospheric values (Frey et al., 2009). This discrepancy has been explained as the NO$_3^-$ incorporating and exchanging isotopically lighter oxygen from local water through a cage effect during re-oxidation of photolytic products (McCabe et al., 2005; Erbland et al., 2015). There is no significant similar reservoir of exchange for nitrogen, and as a result, the net effect of photolytic mass loss and re-oxidation produces so-called "apparent" fractionation constants that are negative for $\delta^{15}N_{NO3}$ and positive for $\delta^{18}O_{NO3}$ and $\Delta^{17}O_{NO3}$ (Röthlisberger et al., 2002; Wolff et al., 2002; Blunier et al., 2005; Grannas et al., 2007; McCabe et al., 2007; e.g., Frey et al., 2009; Winton et al., 2020). As sunlight is rapidly attenuated beneath the snow surface, photolytic loss is restricted to the photic zone (i.e., the 0.1–1.0 m deep zone that light can penetrate and sustain photochemical reactions) and is most pronounced in the uppermost few centimeters of the snowpack (Frey et al., 2009; Zatko et al., 2013; Erbland et al., 2015; Winton et al., 2020).

Although photolysis dominates post-depositional changes to NO$_3^-$, other factors can also play minor roles. Wind can physically mix snow bearing NO$_3^-$ from different seasons or years and blur pre-existing NO$_3^-$ cycles. Additionally, the development and migration of surface features like dunes and sastrugi can result in wildly variable hyperlocal accumulation rates on short timescales (0.5–5 yr) and across very short distances (< 5 m), even if the mean SMB for the broader region stays constant. These phases of erosion and deposition can result in NO$_3^-$ cycles that appear stretched or compressed relative to expectations from regional SMB or even create stratigraphic unconformities with missing periods of deposition (Frezzotti et al., 2002; Scarchilli et al., 2010; Gautier et al., 2016; Picard et al., 2019). NO$_3^-$ volatilization can also be a source of NO$_3^-$ mass loss in Antarctic snow, but it is largely restricted to the warmest coastal regions of Antarctica and is believed to have little isotopic fractionation impact (Erbland et al., 2013; Shi et al., 2019). Finally, downward transport and re-oxidation of photolytic NO$_x$ within the firn may also occur, but as of yet this process is poorly attested and significant impacts appear to be largely restricted to very dry interior sites (SMB < 40 kg m$^{-2}$ a$^{-1}$) (Akers et al., 2022). Once buried beneath the depth where post-depositional processes are active, NO$_3^-$ is assumed to be practically chemically inert and physically immobile (Frey et al., 2009; Erbland et al., 2013; Shi et al., 2015; Noro et al., 2018), aside from volcanic H$_2$SO$_4$-driven NO$_3^-$ displacement with no changes to isotopic compositions (Wolff, 1995; Röthlisberger et al., 2002; Jiang et al., 2019).

The overall impact of these post depositional effects on NO$_3^-$ in Antarctic snow and ice varies strongly depending upon local SMB (Shi et al., 2015, 2019; Akers et al., 2022). At sites with very high SMB, such as near the Antarctic coast, post-depositional effects have little time to alter NO$_3^-$, and the NO$_3^-$ in ice cores should preserves atmospheric NO$_3^-$ values relatively well in a manner following Late Holocene ice core NO$_3^-$ reported from similarly high SMB Greenland (Hastings et al., 2004; Fibiger et al., 2013). For much of drier inland Antarctica, in contrast, it may take 2–10 years for NO$_3^-$ to reach the "archived zone" beneath the range of post-depositional effects, and the combined effects of post-depositional processes here typically overwhelm and obliterate any NO$_3^-$ seasonal cycle variability (Erbland et al., 2013; Shi et al., 2015). Photolytic impacts, in particular, are sensitive to SMB in East Antarctica with a strong linear correlation observed spatially between $\delta^{15}N_{NO3}$ and the reciprocal of SMB (Akers et al., 2022). Changes in insolation, total column ozone, and snow optical properties also can leave imprints on the isotopic values of NO$_3^-$ by affecting the photolytic rate, but the greater photolytic sensitivity to SMB changes tends to overwhelm and obscure their impact (Zatko et al., 2016; Winton et al., 2020; Akers et

al., 2022; Cao et al., 2022; Shi et al., 2022b). Still, these other photolytic factors remain enticing targets for paleoenvironmental reconstruction.

### 3. Methods

We sampled snow for $NO_3^-$ analysis in Nov–Dec 2013 at 23 sites along the CHICTABA traverse (Table 1) from the D85 skiway (70.425° S, 134.146° E, 2848 m a.s.l.) to the Aurora Basin North (ABN) ice core drilling site (71.167° S, 111.367° E,
2689 m a.s.l.) (Figure 2). For each snow sample, 100–600 g of snow were collected into clean sealed plastic bags and stored frozen in clean conditions until the return to Concordia station. All samples were taken upwind of the traverse route to avoid possible contamination. Total snow sampling consisted of 23 "skin layer" samples that collected the top 2–6 mm of loose surface snow, nine "1 m depth layer" samples taken by mixing a 5–10 cm thick layer of snow from 1 m below the surface, and five snow pits sampled in 3 cm increments to depths of 99 cm (P1), 102 cm (P2, P3, P4), or 201 cm (P5) for 202 total pit
samples. Due to the absence of ground-observed SMB values, we used the 35 km grid output from the Modèle Atmosphérique Régional (MAR) version 3.12.1 forced by ERA5 data for the period 1979–2021 (Agosta et al., 2019; Amory et al., 2021). Site-specific SMB values were extracted from the MAR output through bilinear interpolation of the four nearest grid cells, and SMB uncertainties were estimated by comparing model output to known in situ observations (Supplementary Text S1). As the entire transect is located south of the Antarctic Circle, each site experiences extreme seasonal changes in
daylength with a period of 24 hr night in the winter and a period of 24 hr daylight in the summer.

Each snow sample was melted at room temperature in Concordia station, Dome C, Antarctica, and $NO_3^-$ concentrations of the melted samples were determined on aliquots by a colorimetric method with a detection limit of 0.5 ng g$^{-1}$ and precision < 3 % (Frey et al., 2009; Erbland et al., 2013). Melted samples were immediately passed through an anionic exchange resin (Bio-Rad™ AG 1-X8, chloride form), and the resulting trapped $NO_3^-$ eluted with 2 x 5 ml of NaCl 1 M solution. These
concentrated samples were then frozen and shipped to the Institut des Géosciences de l'Environnement (IGE), Grenoble, France, for isotopic analysis. Once re-melted, $NO_3^-$ in these samples was converted to $N_2O$ with a strain of the denitrifying bacteria *Pseudomonas aureofaciens* that lacks the ability to reduce $N_2O$ into $N_2$. The $N_2O$ was thermally decomposed into $O_2$ and $N_2$ on a 900° C gold surface, separated by gas chromatography with a GasBench II™, and oxygen and nitrogen isotopic ratios measured on a Thermo Finnigan™ MAT 253 mass spectrometer (Sigman et al., 2001; Casciotti et al., 2002; Kaiser et
al., 2007; Morin et al., 2009). Isotopic effects from this analysis were corrected using calibration regressions based on standards of international reference materials USGS 32, USGS 34, and USGS 35 processed and analyzed along with each set of samples (Frey et al., 2009; Morin et al., 2009). Standards and samples strictly follow an identical treatment, having the same liquid volume, bacterial culture, and water isotope composition. Isotopic values are reported relative to the $N_2$-Air and VSMOW standard references (Baertschi, 1976; Mariotti, 1983), and the root mean square errors of standards run alongside
our samples over four analytical runs were ±0.7–1.1 ‰ for $\delta^{15}N_{NO3}$, ±0.8–2.3 ‰ for $\delta^{18}O_{NO3}$, and ±0.2–0.4 ‰ for $\Delta^{17}O_{NO3}$. For statistical results reported throughout this paper, uncertainties are given as 95 % confidence intervals unless otherwise stated, and statistical significance is identified as $p$-values < 0.05.

Apparent fractionation constants ($^z\varepsilon_{app}$, where $^{15}\varepsilon = \delta^{15}N_{NO3}$, $^{18}\varepsilon = \delta^{18}O_{NO3}$, and $^{17}E = \Delta^{17}O_{NO3}$) were calculated at all sites through linear regressions of skin layer samples with samples taken at 1 m depth and along the pit profiles. As the site
CHIC-01 did not have a skin layer sample, we extrapolated skin layer values of $\omega(NO_3^-)$, $\delta^{15}N_{NO3}$, $\delta^{18}O_{NO3}$, and $\Delta^{17}O_{NO3}$ from other sites' skin layer data using linear regressions calculated between these variables and site-specific SMB. This extrapolated value was only used in the fractionation constant and pit cycle calculations and otherwise not included in statistical analyses and figures. In line with previous studies (Blunier et al., 2005; Shi et al., 2015), $\varepsilon_{app}$ values are calculated as the slope of a linear regression through Eq. (1):

$\ln R_f = \varepsilon \cdot \ln \omega_f + \ln R_0$        (1)

where $R_0$ and $R_f$ denote isotopic ratios in the initial and remaining $NO_3^-$ and $\omega_f$ denotes the mass fraction of remaining $NO_3^-$. This equation can also be written with delta notation:

$\ln(\delta_f + 1) = \varepsilon \cdot \ln \omega_f + \ln(\delta_0 + 1)$        (2)

where $\delta_0$ and $\delta_f$ denote the desired isotopic species in delta notation versus a chosen standard (e.g., $\delta^{18}O_{NO3}$ vs. VSMOW).

For the subset of skin layer sites that had a paired 1 m depth layer sample, this regression is simple as it only has two points. In the pits, however, the regressions did not capture well the broader multiannual photolytic trend due to the limited number of seasonal cycles recorded per pit and due to the irregular magnitude peaks of the $\omega(NO_3^-)$ cycle, which contributed large outlier points. We therefore created "pseudo-depth layer samples" for each of the five pits to represent an annually-averaged $NO_3^-$ value from below the photic zone. These pseudo-depth layer samples are the $\omega(NO_3^-)$-weighted means of $\omega(NO_3^-)$,

$\delta^{15}N_{NO3}$, $\delta^{18}O_{NO3}$, and $\Delta^{17}O_{NO3}$ for the deepest complete full seasonal cycle observed for P1–P4 and the deepest three complete cycles combined for P5.

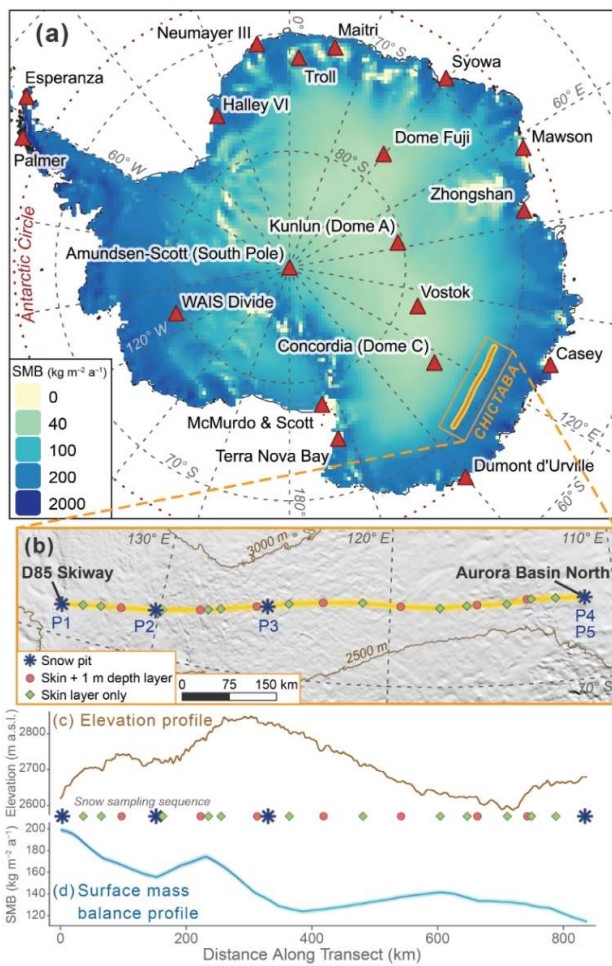

**Figure 2. Maps and environmental profiles of the CHICTABA traverse. (a)** Spatial variability in surface mass balance (SMB) across Antarctica shown by base color shading of MARv3.12.1 output data for the years 2011–2013 (Agosta et al., 2019; Amory et
al., 2021). Major Antarctic stations are labeled (COMNAP, 2017), and the route of the CHICTABA transect is indicated by the orange and yellow line. **(b)** Zoomed map focused on the CHICTABA route (yellow line) overlaid on hillshaded topography with elevation contours shown in brown (Howat et al., 2019). Snow sampling locations along the transect and the sampling method are shown by colored icons with snow pit sites labeled. **(c)** Elevation (Howat et al., 2019) and **(d)** SMB (Agosta et al., 2019; Amory et al., 2021) profiles along CHICTABA starting from the D85 skiway and ending at Aurora Basin North, following the layout of (b),
with the sequence of snow sampling sites along the transect provided. The resolution of the elevation profile reflects the 200 m REMA raster cell-size. The SMB values for the SMB profile were bilinearly interpolated at 1 km intervals from the original 35 km MAR output grid. MAR SMB uncertainty is included on (d) as a shaded zone around the profile line but is difficult to see due to its small size.

**Table 1. Snow sampling site details along the CHICTABA traverse. Elevation is based on the Reference Elevation Model of Antarctica (REMA) (Howat et al., 2019) and surface mass balance (SMB) values are the mean annual SMB output and uncertainty of the MARv3.12.1 for 2011–2013 (Agosta et al., 2019; Amory et al., 2021). Sites are ordered by distance along the traverse from the D85 starting point toward the ABN destination. Note that the sampling dates are not sequential because samples were taken on both the outbound and return trips.**


| Site | Latitude (°) | Longitude (°) | Elevation (m a.s.l.) | SMB 2011–2013 (kg m$^{-2}$ a$^{-1}$) | Sampling date | Pit samples | Skin layer samples | Depth layer samples |
|---|---|---|---|---|---|---|---|---|
| CHIC-01 | −70.431 | 134.138 | 2619 | 198.8 ± 2.2 | 2013-11-30 | P1: 99 cm | | |
| CHIC-02 | −70.500 | 133.264 | 2694 | 188.7 ± 2.2 | 2013-12-26 | | SK23 | |
| CHIC-03 | −70.551 | 132.506 | 2702 | 175.0 ± 2.1 | 2013-12-01 | | SK01 | |
| CHIC-04 | −70.597 | 131.646 | 2740 | 167.2 ± 2.1 | 2013-12-25 | | SK22 | D09 |
| CHIC-05 | −70.675 | 130.172 | 2731 | 155.8 ± 2.0 | 2013-12-02 | P2: 102 cm | SK02 | |
| CHIC-06 | −70.700 | 129.891 | 2718 | 157.5 ± 2.0 | 2013-12-25 | | SK21 | |
| CHIC-07 | −70.804 | 128.282 | 2781 | 172.3 ± 2.1 | 2013-12-24 | | SK20 | D08 |
| CHIC-08 | −70.826 | 127.944 | 2796 | 173.7 ± 2.1 | 2013-12-03 | | SK03 | |
| CHIC-09 | −70.867 | 127.408 | 2824 | 166.9 ± 2.1 | 2013-12-24 | | SK19 | |
| CHIC-10 | −70.979 | 125.863 | 2843 | 140.2 ± 2.0 | 2013-12-23 | | SK18 | D07 |
| CHIC-11 | −70.998 | 125.388 | 2828 | 135.3 ± 2.0 | 2013-12-04 | P3: 102 cm | SK04 | |
| CHIC-12 | −71.070 | 124.474 | 2806 | 126.4 ± 2.0 | 2013-12-23 | | SK17 | |
| CHIC-13 | −71.137 | 122.974 | 2755 | 125.9 ± 2.0 | 2013-12-22 | | SK16 | D06 |
| CHIC-14 | −71.174 | 121.232 | 2713 | 131.2 ± 2.0 | 2013-12-22 | | SK15 | |
| CHIC-15 | −71.145 | 119.534 | 2666 | 137.2 ± 2.1 | 2013-12-21 | | SK14 | D05 |
| CHIC-16 | −71.126 | 117.799 | 2631 | 141.3 ± 2.1 | 2013-12-21 | | SK13 | |
| CHIC-17 | −71.155 | 116.607 | 2623 | 136.4 ± 2.0 | 2013-12-07 | | SK05 | |
| CHIC-18 | −71.165 | 116.151 | 2617 | 133.6 ± 2.0 | 2013-12-20 | | SK12 | D04 |
| CHIC-19 | −71.157 | 114.826 | 2697 | 132.6 ± 2.0 | 2013-12-20 | | SK11 | |
| CHIC-20 | −71.212 | 113.927 | 2638 | 129.8 ± 2.0 | 2013-12-19 | | SK10 | D03 |
| CHIC-21 | −71.210 | 113.740 | 2652 | 129.0 ± 2.0 | 2013-12-08 | | SK06 | |
| CHIC-22 | −71.198 | 112.657 | 2666 | 123.9 ± 2.0 | 2013-12-19 | | SK09 | |
| ABN | −71.167 | 111.367 | 2679 | 114.3 ± 1.9 | 2013-12-12 | P4: 102 cm | | |
| ABN | −71.167 | 111.367 | 2679 | 114.3 ± 1.9 | 2013-12-14 | | SK07 | D01 |
| ABN | −71.167 | 111.367 | 2679 | 114.3 ± 1.9 | 2013-12-17 | P5: 201 cm | SK08 | D02 |

The $\varepsilon_{app}$ values produced in this manner give insight into the isotopic fractionation processes at work, but they also have limitations that are important to recognize. Namely, the most accurate $\varepsilon_{app}$ determinations require many samples taken over a full photic zone profile to compensate for seasonal and environmental variability in $\omega(NO_3^-)$ and $\delta^{15}N_{NO3}$ values (Shi et al., 2015). This is of particular importance at sites where annual snow accumulation is greater than the $NO_3^-$ sampling resolution, as is the case for our CHICTABA samples. Because our 1 m depth samples were taken as the mixed aggregate of a layer only 5–10 cm thick, each 1 m depth sample collects only part of a complete annual $NO_3^-$ cycle. Assuming that the odds of the exact seasonal timing sampled by each 1 m depth sample is stochastic, our individual $\varepsilon_{app}$ values should be viewed as having wide uncertainty with regards to the true site $\varepsilon_{app}$ value, but $\varepsilon_{app}$ values averaged across our dataset should reflect accurate regional $\varepsilon_{app}$ values.

To examine spatial relationships in $NO_3^-$ with SMB along the CHICTABA transect, we calculated linear regressions between $NO_3^-$ variables and local SMB for both skin layer samples and 1 m depth layer samples (including the five pit pseudo-depth layer samples). Following the relationships defined in Akers et al. (2022), regressions were performed as $\omega(NO_3^-)$ or $\ln(\delta_f + 1)$ versus SMB$^{-1}$. The SMB values used in these regressions were the mean annual MAR output for the period 2011–2013 (i.e., the three years preceding sampling). This period was chosen because three years of snowfall at the CHICTABA sites is roughly equal to 1 m of accumulation and compaction. Additional regressions were calculated using the mean annual MAR output for the full data coverage period of 1979–2021 and for the sole year 2013 to determine if the choice of SMB data period substantially affected results. We again assume that any seasonal bias introduced by the 1 m depth sampling technique would be stochastic and that conclusions drawn from observations integrating all sites are generally accurate but admittedly more imprecise than if the individual 1 m depth samples had integrated full annual cycles. Statistical calculations and figure production were performed using the R programming language with packages *tidyverse, lubridate, RColorBrewer, gridExtra, cowplot, raster, rts, ncdf4, RMisc,* and *HMisc.* QGIS was used for spatial analyses and map creation using data produced here or cited in image captions with Adobe Illustrator used for finalization of figures.

## 4. Results

In total, 234 individual snow samples were analyzed for $\omega(NO_3^-)$ and $NO_3^-$ isotopic ratios (Figure 3). Skin layer samples have the highest $\omega(NO_3^-)$, with values from 124 to 501 ng g$^{-1}$, and 1 m depth layer samples have lower $\omega(NO_3^-)$ between 49 and 97 ng g$^{-1}$. Each pit has a wide range of $\omega(NO_3^-)$ values that fall between the values observed in the skin layer and at 1 m depth (Figure 3a). Skin layer samples have $\delta^{15}N_{NO3}$ values that are largely below 0 ‰ (mean: $-8.9 \pm 3.3$ ‰) and within the range observed in atmospheric $NO_3^-$ (Figure 1). In contrast, nearly all the $\delta^{15}N_{NO3}$ values from the 1 m depth layer (mean: $+46.1 \pm 12.3$ ‰) and pit samples (mean: $+36.0 \pm 3.1$ ‰) are much higher than the skin layer (Figure 3b). Values of $\delta^{18}O_{NO3}$ and $\Delta^{17}O_{NO3}$ are broadly similar across all sample groups ($\delta^{18}O_{NO3}$ all samples mean: $+70.7 \pm 1.4$ ‰, $\Delta^{17}O_{NO3}$ all samples mean: $+30.9 \pm 0.5$ ‰), but drier pit sites (i.e., P3, P4, and P5) have somewhat lower values (Figure 3c–d). For both $\omega(NO_3^-)$ and $\delta^{15}N_{NO3}$, the mean values between the skin layer and 1 m depth layer sample sets are strongly and significantly differentiated (Mann-Whitney U test, $p \ll 0.01$), while the differences between skin and 1 m depth layer samples for both $\delta^{18}O_{NO3}$ and $\Delta^{17}O_{NO3}$ are less clear but still statistically significant at $p = 0.01$ and 0.04, respectively (Mann-Whitney U test).

Data from the pits ($n = 202$ pit $+ 5$ site skin layer) show cyclical patterns in $\omega(NO_3^-)$ and isotopic values as well as linear trends across the entire depths of the pits (Figure 4, Supplementary Table S1). The pits have 2–2.5 cycles in the top 100 cm with drier sites containing more cycles per unit depth. For the deeper P5, we observe five complete cycles over the total 201 cm depth. Linear regressions of $NO_3^-$ variables with depth reveal that $\omega(NO_3^-)$ has statistically significant negative slopes at P3–P5 ($p < 0.01$) while $\delta^{15}N_{NO3}$ has a significant positive slope only at P5 ($p = 0.02$). Both $\delta^{18}O_{NO3}$ and $\Delta^{17}O_{NO3}$ have statistically significant negative slopes at all pits except P1 ($p < 0.01$). In the absence of other supplemental geochemical

data, the residuals of the $\Delta^{17}O_{NO3}$ regression with depth were used to identify seasonal cycles with positive residuals representing colder months and negative residuals representing warmer months (Figure 4). This seasonal identification is based on $NO_3^-$ monitoring data from Dome C (Figure 1) and previously reported seasonal $\Delta^{17}O_{NO3}$ cycles linked to snow $\delta^{18}O$ variability in a snow pit (Shi et al., 2015).

We investigated how the cycle timing of $NO_3^-$ variables were interrelated by correlating values after we removed linear
trends with depth (i.e., we correlated the residuals of the linear regressions). Values for $\delta^{18}O_{NO3}$ and $\Delta^{17}O_{NO3}$ values are well-correlated ($r = +0.72$, $p < 0.01$), as is typically observed for $NO_3^-$. The $\omega(NO_3^-)$ has a moderate negative correlation with $\delta^{18}O_{NO3}$ ($r = -0.34$, $p < 0.01$) and a weak negative correlation with $\Delta^{17}O_{NO3}$ ($r = -0.16$, $p = 0.03$), while $\omega(NO_3^-)$ and $\delta^{15}N_{NO3}$ do not have a statistically significant relationship ($r = -0.11$, $p = 0.16$). Although $\delta^{15}N_{NO3}$ has fairly strong positive correlation with $\Delta^{17}O_{NO3}$ ($r = +0.51$, $p < 0.001$), there is no significant relationship between $\delta^{15}N_{NO3}$ and $\delta^{18}O_{NO3}$ ($r = -0.06$,
$p = 0.43$). This difference in correlation strength seems unusual since $\delta^{18}O_{NO3}$ and $\Delta^{17}O_{NO3}$ are so strongly correlated, but it appears to arise because the $\delta^{18}O_{NO3}$ cycle is slightly more irregular and offset from the $\delta^{15}N_{NO3}$ cycle compared with the $\Delta^{17}O_{NO3}$ values (Figure 4). Additionally, $\Delta^{17}O_{NO3}$ values tend to peak higher than $\delta^{18}O_{NO3}$ values when coinciding with the highest $\delta^{15}N_{NO3}$ values (e.g., P2: 75 cm, P3: 35 cm, P4: 55 cm), and these shared extreme values promote a stronger correlation. The reason for these small differences between $\delta^{18}O_{NO3}$ and $\Delta^{17}O_{NO3}$ is not presently clear but may be due to
$\delta^{18}O_{NO3}$ values being theoretically directly affected by photolytic mass loss while $\Delta^{17}O_{NO3}$ is not. Unfortunately, the impact of a theoretical fractionation of oxygen isotopes by photolytic mass loss is poorly constrained due to competing effects from oxygen atomic exchange during $NO_3^-$ re-oxidation, which we examine in more detail later.

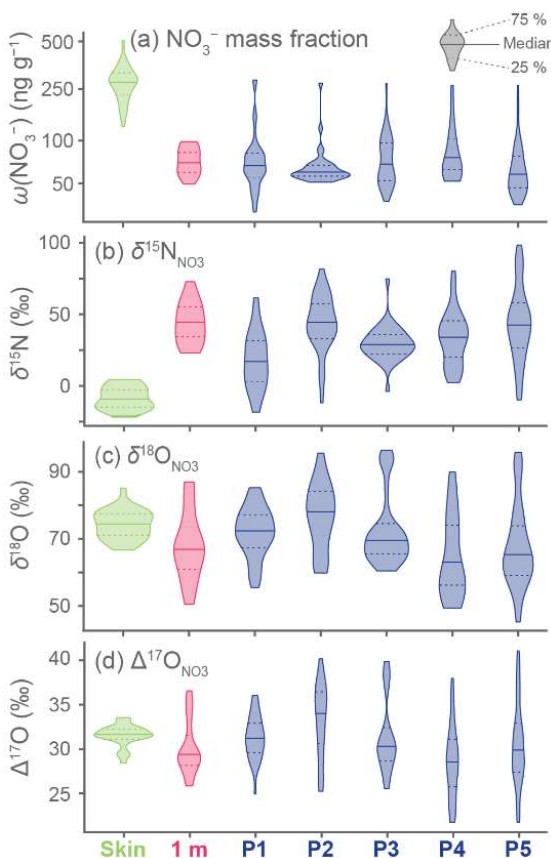

**Figure 3. Violin plots showing the distributions of $NO_3^-$ analytical results. Samples in each subplot are grouped and colored by**
**sampling method: skin layer (green), 1 m depth layer (pink), or pits P1–P5 (blue). Data are plotted so that the total area of distribution is equivalent between groups, regardless of sample count. The median value per group is shown by a solid horizontal line, while the 25th and 75th percentiles are shown by dashed horizontal lines. Note that the y-axis for $\omega(NO_3^-)$ (a) is log-transformed to better display the much higher $NO_3^-$ concentrations in the skin layer samples relative to other sample groups.**


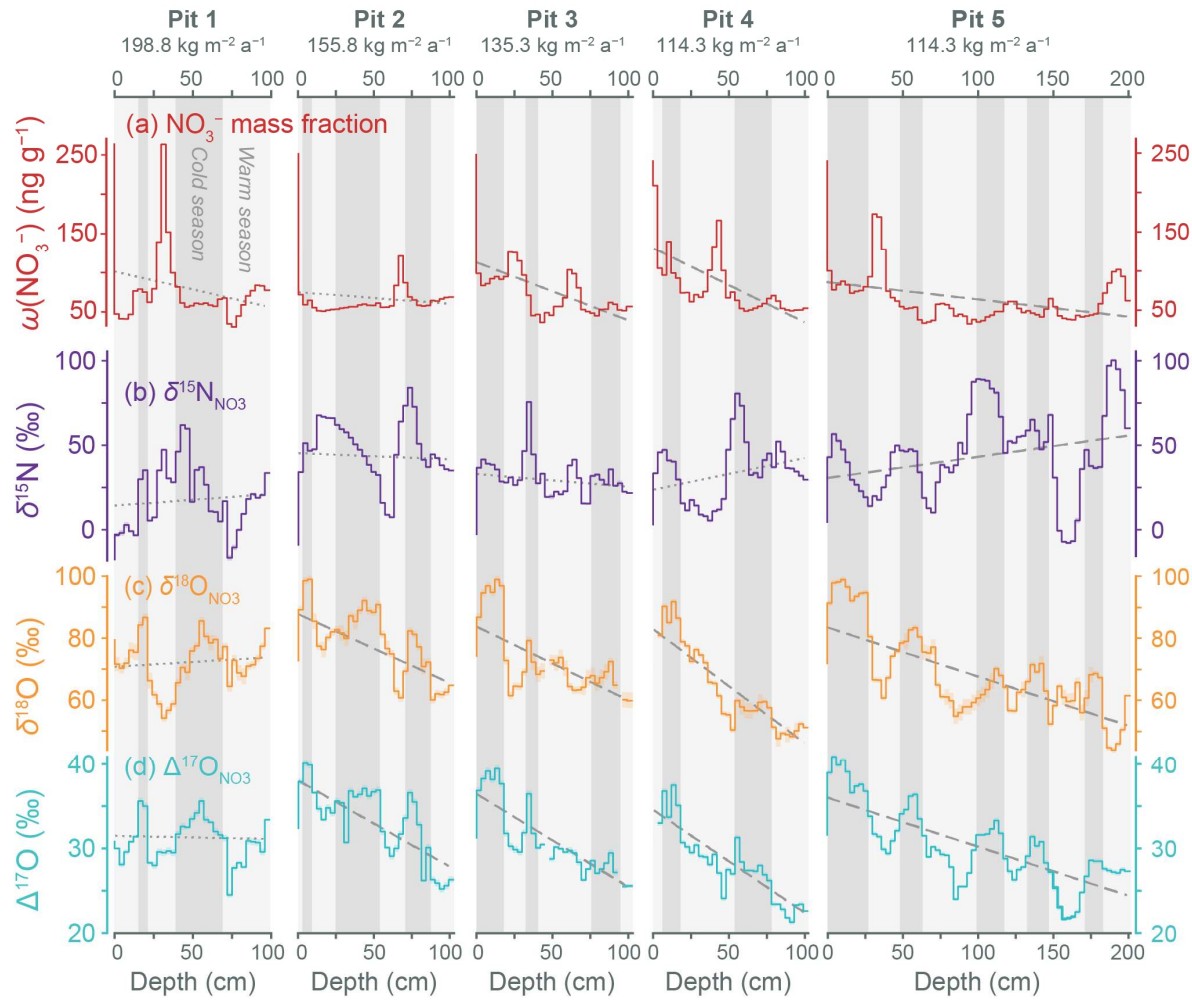

**Figure 4. Changes in ω(NO₃⁻) (a) and NO₃⁻ isotopic values (b–d) with snow depth for five pits sampled along the CHICTABA traverse. The modeled surface mass balances (Agosta et al., 2019; Amory et al., 2021) for different pit sites are given at the top of each plot. Dashed and dotted gray lines show a linear regression (variable vs. depth) fitted to each set of data (Supplementary Table S1). Dashed lines represent regressions whose f-statistic p-value < 0.05, and dotted lines represent regressions whose f-**
**statistic p-value ≥ 0.05. Gray shaded backgrounds indicate inferred seasonal cycles (darker = colder months of ~May–Oct, lighter = warmer months of ~Nov–Apr) based primarily on when residuals of the Δ¹⁷O_{NO3} regression are positive (i.e., Δ¹⁷O_{NO3} peaks). Measurement uncertainties in isotopic values are displayed as colored shaded zones around the stepped lines but are too small to be visible on most data.**

Across the full 1979–2021 dataset, we find interannual SMB variability to be very high, but the spatial pattern of variability is consistent year to year (Supplementary Figure S1, Supplementary Table S2). Model uncertainties in annual SMB values were estimated at ±1.6–2.5 kg m⁻² a⁻¹ by comparing model output to in situ observations (Supplementary Text S1). For the period 2011–2013, mean SMB values at sampling sites ranged from a high of 198.8±2.2 kg m⁻² a⁻¹ at the D85/CHIC-01 transect start to a low of 114.3±1.9 kg m⁻² a⁻¹ at the ending ABN site (Figure 2d). Regressions performed with the 1979–

2021 and 2013 SMB datasets produce very similar results to those of the 2011–2013 dataset (Supplementary Figure S2, Supplementary Table S3). Generally, slope values for the 1979–2021 and 2013 regressions have greater magnitude than 2011–2013 because the overall range in SMB values along CHICTABA in 2011–2013 was greater than during the other two time periods.

Only some of the NO₃⁻ variables have statistically significant linear relationships with the 2011–2013 SMB⁻¹ values (Figure

5, Table 2). The δ¹⁵N_{NO3} values decrease with higher SMB in both the skin layer and 1 m depth layer samples (Figure 5b),

but only the skin layer regression has a statistically significant *f-statistic* ($p < 0.01$, n = 23). For oxygen isotopes, only $\delta^{18}O_{NO3}$ in the 1 m depth samples has a statistically significant regression with $SMB^{-1}$ (*f-statistic* $p = 0.04$, n = 14) with higher isotopic values associated with greater snow accumulation (Figure 5c). The $\omega(NO_3^-)$ and $\Delta^{17}O_{NO3}$ values do not have statistically significant relationships with $SMB^{-1}$ in either the skin layer or 1 m depth layer (Figure 5a, d). Although not reaching our defined level of statistical significance, we note that nonzero slope relationships with $SMB^{-1}$ can also be observed in $\delta^{18}O_{NO3}$ in the skin layer and in $\delta^{15}N_{NO3}$ and $\Delta^{17}O_{NO3}$ at 1 m depth. Comparing these results to regressions calculated with SMB from 1979–2021 and from 2013, we find that statistically significant variables are the same across all three time periods with the exception that the 1 m depth $\Delta^{17}O_{NO3}$ regression reaches significance with both 1979–2021 (*f-statistic* $p = 0.04$, n = 14) and 2013 (*f-statistic* $p = 0.05$, n = 14) SMB values but not with 2011–2013 values (Supplementary Table S3).

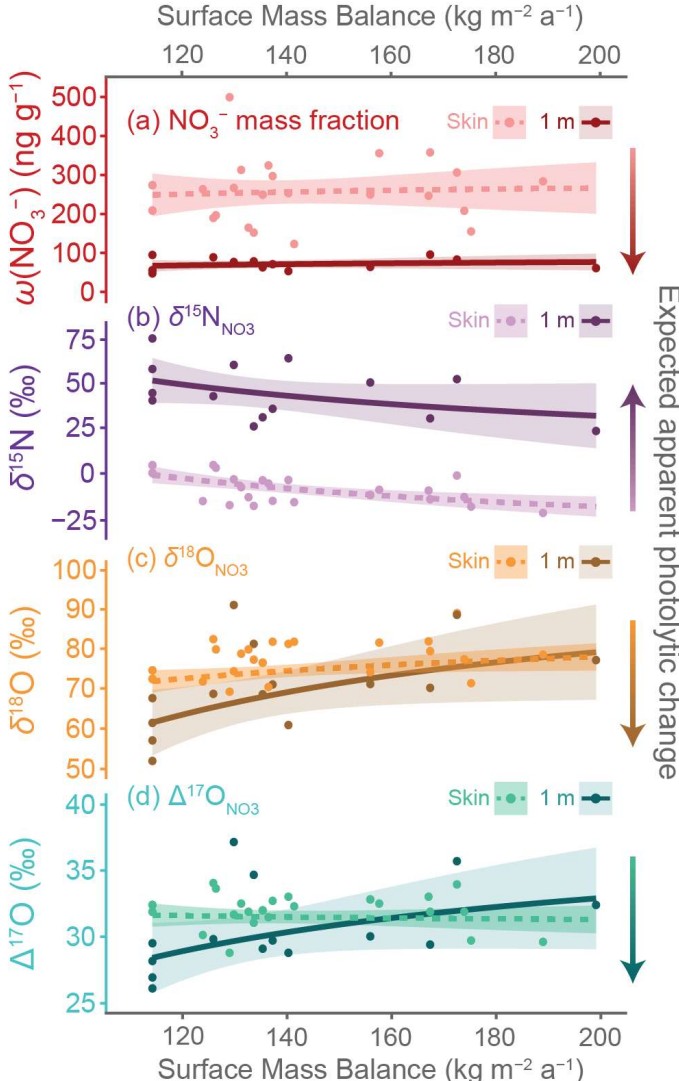

**Figure 5. Spatial relationships between nitrate variables and site surface mass balance (SMB). Linear regressions of (a) $\omega(NO_3^-)$ and (b–d) $\ln(NO_3^-$ isotopic variable + 1) versus $SMB^{-1}$ (Agosta et al., 2019) are shown by dashed (skin layer) and solid (1 m depth layer) lines with 95 % confidence intervals of the regression shown by shaded zones. The SMB values are mean annual values for 2011–2013 from MARv3.12.1 (Agosta et al., 2019; Amory et al., 2021). Individual points represent individual samples. The direction of expected changes to $NO_3^-$ variables due to photolysis and associated re-oxidation is indicated by colored arrows. Coefficients and statistics for displayed regressions are given in Table 2.**

Table 2. Coefficients and statistics for the linear regressions of $NO_3^-$ variables versus local site $SMB^{-1}$, with skin layer and 1 m depth layer samples separately analyzed. Coefficient values are given with ±1 standard error. Values of SMB used in regressions are the mean annual output of MARv3.12.1 forced with ERA5 data for the years 2011–2013 (Agosta et al., 2019; Amory et al., 2021). Regressions with statistically significant $f$-statistic ($p < 0.05$) values are bolded.

| Skin layer samples | | | | |
|---|---|---|---|---|
| *Variable* | *Slope* | *Intercept* | *F-statistic p-value* | *$r^2$* |
| | *(ng g$^{-1}$ · kg m$^{-2}$ a$^{-1}$ or kg m$^{-2}$ a$^{-1}$)* | *(ng g$^{-1}$ or unitless)* | | |
| $\omega(NO_3^-)$ | −4227 ± 18373 | 290 ± 131 | 0.82 | 0.00 |
| **ln($\delta^{15}N_{NO3}$ + 1)** | **4.0 ± 1.5** | **0.0 ± 0.0** | **0.01** | **0.26** |
| ln($\delta^{18}O_{NO3}$ + 1) | −1.4 ± 0.9 | 0.1 ± 0.0 | 0.11 | 0.11 |
| ln($\Delta^{17}O_{NO3}$ + 1) | 0.1 ± 0.3 | 0.0 ± 0.0 | 0.71 | 0.01 |
| 1 m depth layer samples | | | | |
| *Variable* | *Slope* | *Intercept* | *F-statistic p-value* | *$r^2$* |
| | *(ng g$^{-1}$ · kg m$^{-2}$ a$^{-1}$ or kg m$^{-2}$ a$^{-1}$)* | *(ng g$^{-1}$ or unitless)* | | |
| $\omega(NO_3^-)$ | −2626 ± 3745 | 91 ± 28 | 0.50 | 0.04 |
| ln($\delta^{15}N_{NO3}$ + 1) | 5.1 ± 3.1 | 0.0 ± 0.0 | 0.12 | 0.19 |
| **ln($\delta^{18}O_{NO3}$ + 1)** | **−4.4 ± 2.0** | **0.1 ± 0.0** | **0.04** | **0.30** |
| ln($\Delta^{17}O_{NO3}$ + 1) | −1.2 ± 0.7 | 0.0 ± 0.0 | 0.10 | 0.21 |

Table 3. Apparent $NO_3^-$ isotopic fractionation constants for sites along the CHICTABA traverse. Values for $\delta^{15}N_{NO3}$ ($^{15}\varepsilon_{app}$), $\delta^{18}O_{NO3}$ ($^{18}\varepsilon_{app}$), and $\Delta^{17}O_{NO3}$ ($^{17}E_{app}$) were calculated from the paired skin layer and 1 m depth samples at individual sites. The MAR-estimated surface mass balance (SMB) (Agosta et al., 2019; Amory et al., 2021) is provided for each site for reference, and further site information is given in Table 1. For the five pit samples (P1–P5), a pseudo-depth layer sample was calculated by weight-averaging samples representing at least one full annual cycle and paired with a skin layer sample taken from the same site. Note that the site ABN was sampled four separate times within a five-day period.

| *Site* | *$^{15}\varepsilon_{app}$ (‰)* | *$^{18}\varepsilon_{app}$ (‰)* | *$^{17}E_{app}$ (‰)* | *SMB 2011–2013 (kg m$^{-2}$ a$^{-1}$)* |
|---|---|---|---|---|
| CHIC-01 (P1) | −28.6 | 3.1 | −0.4 | 198.8 ± 2.2 |
| CHIC-04 | −33.2 | 6.1 | 1.8 | 167.2 ± 2.1 |
| CHIC-05 (P2) | −44.5 | 2.0 | 2.0 | 155.8 ± 2.0 |
| CHIC-07 | −39.4 | 0.3 | −1.3 | 172.3 ± 2.1 |
| CHIC-10 | −41.7 | 11.6 | 2.6 | 140.2 ± 2.0 |
| CHIC-11 (P3) | −24.8 | 5.1 | 2.0 | 135.3 ± 2.0 |
| CHIC-13 | −48.6 | 15.9 | 5.3 | 125.9 ± 2.0 |
| CHIC-15 | −34.9 | 6.6 | 2.0 | 137.2 ± 2.1 |
| CHIC-18 | −65.6 | −5.3 | −5.2 | 133.6 ± 2.0 |
| CHIC-20 | −48.8 | −11.7 | −4.2 | 129.8 ± 2.0 |
| ABN (P4) | −29.3 | 10.4 | 4.6 | 114.3 ± 1.9 |
| ABN (P5) | −34.4 | 7.3 | 2.2 | 114.3 ± 1.9 |
| ABN (12-Dec) | −45.3 | 12.4 | 3.5 | 114.3 ± 1.9 |
| ABN (17-Dec) | −36.6 | 5.8 | 2.1 | 114.3 ± 1.9 |
| *Mean ± 95 % CI* | *−39.7 ± 6.1* | *5.0 ± 4.2* | *1.2 ± 1.7* | |
| *Median* | *−39.4* | *5.8* | *2.0* | |

Apparent fractionation constants for each of the isotopic ratios have high variability across all sites (Table 3). This variability is likely due in part to the sampling methodology where the skin layer sample will have summer values (as we collected it in summer), but the 1 m depth samples reflect a random sampling from a buried seasonal cycle. While this reduces the precision of our overall $\varepsilon_{app}$ estimate, general conclusions can be drawn from the range of $\varepsilon_{app}$ values as well as their measures of central tendency. The $^{15}\varepsilon_{app}$ values are all negative and range between –65.6 ‰ and –24.8 ‰, with a mean value of –39.7 ± 6.1 ‰. Fractionation constants for oxygen isotopes are positive except at two sites, but smaller in magnitude than that of the nitrogen isotopes: $^{18}\varepsilon_{app}$ values range between −11.7 ‰ and +15.9 ‰ (mean: +5.0 ± 4.2 ‰) and $^{17}E_{app}$ range between −5.1 ‰ and +5.3 ‰ (mean: +1.2 ± 1.7 ‰). Fractionation constants do not have statistically significant linear regressions with either SMB or SMB$^{-1}$.

## 5. Discussion

### 5.1 Photolytic impacts observed in skin layer and 1 m depth samples

Our data reveal evidence of photolytic changes to $NO_3^-$ in the photochemically active zone of the snowpack. The mean $\delta^{15}N_{NO3}$ of skin layer samples (−8.9 ± 3.3 ‰) is within the typical seasonal range (≈−40 to +20 ‰) observed in atmospheric $NO_3^-$ at both coastal and interior Antarctic stations (Savarino et al., 2007; Frey et al., 2009; Erbland et al., 2013; Winton et al., 2020; Shi et al., 2022a), suggesting that the skin layer $NO_3^-$ is recently deposited from the atmosphere and has experienced little to no photolytic effects. In contrast, the $\delta^{15}N_{NO3}$ values at 1 m depth are 49 ± 11 ‰ higher on average than the skin layer $\delta^{15}N_{NO3}$. This increase, combined with the average 71 ± 9 % ng g$^{-1}$ drop in $\omega(NO_3^-)$ from the skin layer to the 1 m depth, strongly points to substantial photolytic mass loss (Savarino et al., 2007; Frey et al., 2009; Meusinger et al., 2014; Zatko et al., 2016). As further support, the range of $^{15}\varepsilon_{app}$ values (−65.6 ‰ to −24.8 ‰) at the CHICTABA sites is comparable to both modeled and field-observed values previously reported for photolytic fractionation across interior Antarctic transects (−76.8 ‰ to −31.5 ‰) (Frey et al., 2009; Erbland et al., 2013; Berhanu et al., 2015; Shi et al., 2015). While HNO$_3$ volatilization is not excluded as a minor source of mass loss at some sites due to the wide range in fractionation factors, photolytic mass loss alone can explain the observed findings without needing to invoke additional, non-fractionating mass loss from volatilization.

Photolysis-related impacts on oxygen isotopes are also present but more subtle. Our apparent isotopic fractionation factors for $^{18}\varepsilon_{app}$ and $^{17}E_{app}$ are comparable to oxygen fractionation factors reported in other photolysis studies (Frey et al., 2009; Erbland et al., 2013; Berhanu et al., 2015; Shi et al., 2015) that are a combination of effects from both photolytic mass loss fractionation and oxygen exchange due to a cage effect during re-oxidation of photolytic products. As is expected from photolysis and the resulting $NO_3^-$ re-oxidation, mean values for $\delta^{18}O_{NO3}$ and $\Delta^{17}O_{NO3}$ along the CHICTABA transect are lower at 1 m than in the skin layer. However, the difference between the mean skin layer and 1 m depth values is much smaller than observed in $\delta^{15}N_{NO3}$. This is reflected in how the apparent fractionation factors for $\delta^{18}O_{NO3}$ and $\Delta^{17}O_{NO3}$ are much closer to zero than the apparent fractionation factor for $\delta^{15}N_{NO3}$ (Frey et al., 2009; Erbland et al., 2013; Shi et al., 2015), and thus photolytic mass loss did not result in as large of a change in isotopic value for oxygen as for nitrogen.

### 5.2 Annual nitrate cycle and photolytic evidence observed in pit samples

We interpret the cyclical variability of $\omega(NO_3^-)$ and $NO_3^-$ in the depth profiles of the CHICTABA snow pits (Figure 4) as a relic of the annual cycles observed in atmospheric and skin layer $NO_3^-$ (Figure 1) that has been partially preserved through $NO_3^-$ deposition and initial burial. The cycles in $\delta^{18}O_{NO3}$ and $\Delta^{17}O_{NO3}$ are clear and well-synchronized in each pit, which allows us to differentiate between the winter darkness season (peaks) and summer sunlit season (troughs) (Savarino et al.,

2007; Frey et al., 2009; Shi et al., 2015). Peaks in $\omega(NO_3^-)$ generally coincide with the summer minima in the oxygen isotopic cycles due to enhanced deposition of recycled $NO_3^-$ (Figure 4a), as similarly observed in $NO_3^-$ monitoring at Dome C (Figure 1) and in three snow pits reported in a previous study (Shi et al., 2015). However, a few minor peaks observed in winter (e.g., in P1 and P4) could represent $NO_3^-$ deposition from stratospheric denitrification. Additionally, the annual $\omega(NO_3^-)$ peak corresponding to summer 2012–2013 (i.e., the summer before sampling occurred) is particularly large relative to other $\omega(NO_3^-)$ peaks in most pits. This may represent a particularly heavy local $NO_3^-$ deposition that year, although atmospheric and skin layer $NO_3^-$ monitoring at Dome C captured no unusually high $NO_3^-$ at that time (Erbland et al., 2013; Winton et al., 2020). Overall, the range and cycles in $\omega(NO_3^-)$ values observed in these CHICTABA pits are similar to those reported from pits with similar SMB on a transect from Zhongshan station to Dome A (Shi et al., 2018b).

Following that each complete oxygen isotopic cycle is equivalent to one year, the pits cover roughly 2–3.5 years of snow accumulation in the top 100 cm, with five years of accumulation at the 201 cm deep P5. This accumulation is similar to rough estimates (P1: 2.0 yr; P2: 2.5 yr; P3: 2.9 yr; P4: 3.4 yr; P5: 7.0 yr) calculated from modeled SMB for 2011–2013 and snow density profiles taken from two shallow cores along the transect (where 1 m snow depth = 38.9 cm water equivalent and 2.25 m snow depth = 90.4 cm water equivalent). Differences between the modeled estimates and the dating from $NO_3^-$ oxygen isotope cycles could be due to interannual snowfall variability, surface roughness, and/or localized differences in snow density profiles. A surface roughness effect may explain the exceptionally broad $\delta^{15}N_{NO3}$ peak and lack of $\omega(NO_3^-)$ spike in the upper 50 cm of P2 as a localized high rate of drifted snow accumulation that "stretched" the typical cycle frequency. Otherwise, the general regularity of the isotopic cycles suggests that limited physical mixing or snow layer disturbance occurred after initial deposition.

Although photolysis only occurs during sunlit periods, it affects $NO_3^-$ deposited in all seasons. For the pit data, the cyclical patterns of $\delta^{15}N_{NO3}$ and $\delta^{18}O_{NO3}$ are offset 10–80 ‰ higher and 5–15 ‰ lower, respectively, compared to the mean seasonal cycle values reported from the skin layer at Dome C (Erbland et al., 2013). Because it takes over two years for newly deposited $NO_3^-$ to be buried below 1 m along the CHICTABA traverse, $NO_3^-$ that is deposited in winter darkness will still be exposed to summer sunlight and partially photolyzed before being fully buried below the photic zone. We also find that $NO_3^-$ deposited in the late winter and early spring has the greatest $\delta^{15}N_{NO3}$ increase relative to its corresponding seasonal skin layer values, with pit $\delta^{15}N_{NO3}$ values of 50–100 ‰ compared to skin layer mean values of 10–30 ‰. The $^{15}N$ enrichment maximum at this time can be expected because the $NO_3^-$ deposited during late winter and early spring will typically have been buried perhaps 5–20 cm beneath the surface by the time intense summer insolation returns. At this depth, the late winter/early spring $NO_3^-$ is still shallow enough to be readily photolyzed, but also deep enough that newly recycled, isotopically light $NO_3^-$ deposited onto the surface will not be mixed in.

The oxygen isotope values have very clear negative trends with depth in P2–P5 (Figure 4c–d). While the $\delta^{18}O_{NO3}$ and $\Delta^{17}O_{NO3}$ value ranges in the first 25 cm are similar to skin layer values observed at Dome C (Figure 1g–h), the pit values at 75–201 cm are 20–40 ‰ lower for $\delta^{18}O_{NO3}$ and 8–14 ‰ lower for $\Delta^{17}O_{NO3}$ than the Dome C skin layer. This agrees with previous observations where increased photolysis and its resulting oxygen exchange during $NO_3^-$ re-oxidation produce lower oxygen isotopic ratios (Erbland et al., 2013; Shi et al., 2015). However, it is notable that only P4 and P5 have visibly increasing $\delta^{15}N_{NO3}$ trends with depth as would be expected from cumulative photolytic mass loss. This suggests that substantial oxygen exchange may be occurring regardless of photolytic mass loss, perhaps due to photolytic $NO_x$ being produced and re-oxidized in place without the ventilated transport that leads to mass loss to the atmosphere. In this case of in situ photolysis and re-oxidation, no isotopic effect of photolysis would be observed in nitrogen, but there could be an isotopic change in oxygen due to the chance of an atomic exchange with the local snow grain matrix.

Possible oxygen isotopic changes not triggered by photolysis must also be considered. We expect photolysis to drive the greatest rate of isotopic change in the uppermost depths where radiation is strongest and increasingly less change toward the bottom of the photic zone. We observe this in the $\delta^{15}N_{NO3}$ where $\delta^{15}N_{NO3}$ values greatly increase between each skin layer and ≈6–9 cm depth, even for the P2 and P3 pits where no clear additional photolytic change is present beneath this uppermost zone (Figure 4b). In contrast, the oxygen isotopes have a remarkably consistent rate of isotopic change with depth for P2–P5 (Figure 4c–d). Competition between photolytic mass loss fractionation and oxygen exchange isotopic effects is discussed in the following section as one possible explanation for this difference between nitrogen and oxygen profiles. However, the $\delta^{18}O_{NO3}$ and $\Delta^{17}O_{NO3}$ values in P5 continue to decline steadily from 100–201 cm. These depths are well beneath the photic zone, and therefore the $NO_3^-$ should be isotopically stable. No current mechanism in our current understanding of Antarctic $NO_3^-$ dynamics has been described for oxygen isotopic changes in the snowpack without photolysis, and it is difficult to make strong hypotheses or conclusions at this time in the absence of deeper and/or replicated pits. Further and more extensive field observations will be needed to clarify this uncertainty.

The relative timing of isotopic cycles in the pits has some small but important differences from the cycles observed in the atmosphere and skin layer at Dome C. As best seen in the P2–P5 pits, the $\delta^{15}N_{NO3}$ cycle generally aligns in phase with oxygen isotopes, but with a slight offset so that the $\delta^{15}N_{NO3}$ maxima and minima are 0–10 cm shallower (~0–3.5 months later) than the corresponding oxygen isotope cycles (Figure 4b–d). The delayed $\delta^{15}N_{NO3}$ minima, in particular, is unexpected because the early summer $\delta^{15}N_{NO3}$ minima in atmospheric and skin layer $NO_3^-$ precedes the mid-summer minima in oxygen isotopes by 1–2 months (Figure 1b–d) (Savarino et al., 2007; Erbland et al., 2013; Winton et al., 2020). A similar "delayed" relationship between $\delta^{15}N_{NO3}$ and $\delta^{18}O_{NO3}$ can be observed in three snow pits sampled from the wetter section of the Zhongshan to Dome A traverse route (Shi et al., 2015), suggesting that this phenomenon is not unique to CHICTABA and may be typical for intermediate SMB regions of Antarctica.

This discrepancy between observations in snow pits versus the observations in the atmosphere and skin layer may be explained by the seasonality of photolytic loss (Figure 6). The early summer atmospheric $\delta^{15}N_{NO3}$ minima is due to the photolytic production and subsequent re-oxidation of $NO_x$ with low $\delta^{15}N$ from the snowpack $NO_3^-$, and the skin layer $NO_3^-$ shares a similarly timed $\delta^{15}N_{NO3}$ minima as the re-oxidized $NO_3^-$ is deposited back onto the surface (Figure 1). However, as this skin layer $NO_3^-$ is buried by additional snow, it will be exposed to sunlight in the photic zone for the entire summer season with subsequent photolytic losses and an increase in $\delta^{15}N_{NO3}$ values. In contrast, while $NO_3^-$ deposited toward the end of summer may not initially have $\delta^{15}N_{NO3}$ values as low as in early summer, this $NO_3^-$ will experience far less photolytic-inducing radiation before winter darkness and will likely be buried and protected relatively deep in the photic zone before the next summer begins. In this manner, the late summer $\delta^{15}N_{NO3}$ values could end up as the lowest $\delta^{15}N_{NO3}$ values simply because they are photolytically elevated the least from initial atmospheric values. Likewise, the minimum values in pit oxygen isotope cycles may be shifted slightly earlier in the summer because re-oxidation of photolytic products lowers $\delta^{18}O_{NO3}$ and $\Delta^{17}O_{NO3}$ values through oxygen atomic exchange. Thus, we would observe the oxygen isotopic minima occurring before the nitrogen isotopic minima in the pit profiles, despite the atmospheric and skin layer cycles not exhibiting this pattern.

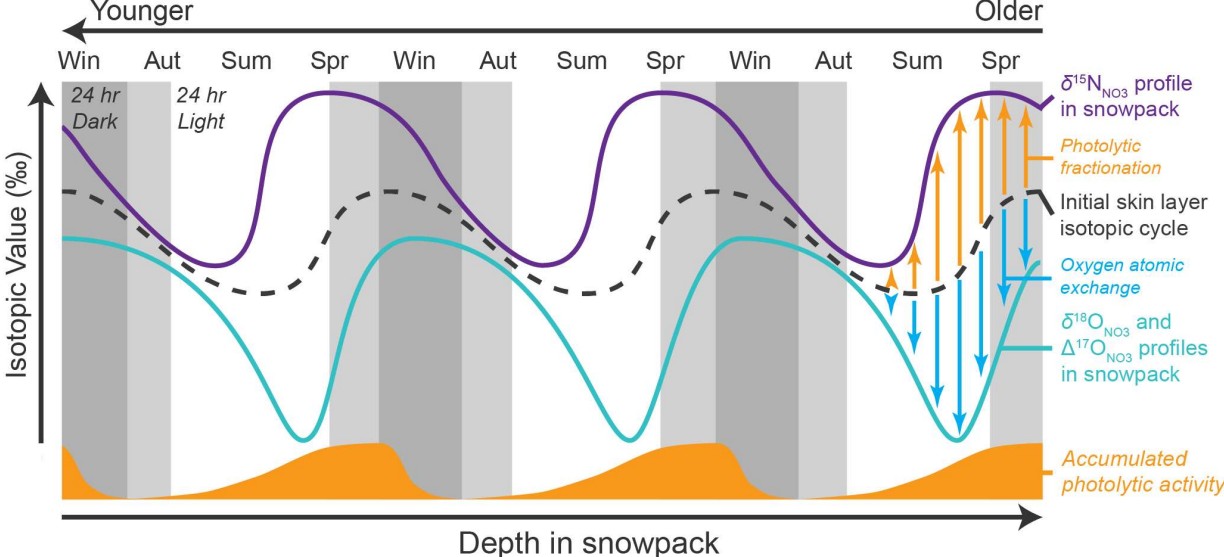

**Figure 6. The proposed mechanism for observed isotopic cycle offsets in snow pit NO$_3^-$. Even if $\delta^{15}N_{NO3}$, $\delta^{18}O_{NO3}$, and $\Delta^{17}O_{NO3}$ have synchronous seasonal isotopic cycles when deposited in the skin layer (black dashed line), post-depositional photolysis will skew $\delta^{15}N_{NO3}$ values (purple solid line) differently than $\delta^{18}O_{NO3}$ and $\Delta^{17}O_{NO3}$ values (teal solid line). Photolysis increases isotopic values for nitrogen due to photolytic fractionation (orange arrows) but decreases values for oxygen due to oxygen atomic exchange (blue**

**arrows). Because the typical amount of photolytic activity experienced by NO$_3^-$ deposited on the snow surface (orange solid curve) also changes seasonally in a cycle not aligned with the skin layer isotopic value cycles, photolysis will enhance or subdue the existing skin layer isotopic cycle differently for $\delta^{15}N_{NO3}$ than for $\delta^{18}O_{NO3}$ and $\Delta^{17}O_{NO3}$. This produces the observed cyclical offsets between nitrogen and oxygen isotopes (Figure 4), even if the magnitude of isotopic value changes due to photolysis (i.e., the size of the arrows) is the same for all isotopic species at a given point in the cycle.**

**5.3 Links between $\delta^{15}N_{NO3}$ and SMB**

The linear relationships between NO$_3^-$ variables and SMB (Figure 5) match what is expected based on photolysis-dominated NO$_3^-$ dynamics on the East Antarctic Plateau. While not all the regressions are statistically significant at $p < 0.05$, their combined evidence supports increased photolysis with lower SMB. At drier sites, NO$_3^-$ will remain within the photic zone for a longer period due to slower snow accumulation, and as a result the NO$_3^-$ will experience more photolysis before being

buried in the archived zone (Akers et al., 2022). For the 1 m depth layer samples, $\delta^{15}N_{NO3}$ values increase while $\delta^{18}O_{NO3}$ and $\Delta^{17}O_{NO3}$ values decrease with lower SMB (Figure 5b–d), which reflects the negative apparent isotopic fractionation factor of nitrogen with NO$_3^-$ photolysis and the positive apparent fractionation factors for oxygen (Erbland et al., 2013; Shi et al., 2015).

An improved sampling method for the 1 m depth samples might produce stronger and more precise linear regressions with

505 SMB$^{-1}$ for all isotopic ratios. Each seasonal isotopic cycle typically covers 30–50 cm depth in the upper snowpack as observed in the pit records (Figure 4). However, each 1 m depth sample taken along the CHICTABA transect likely represents only part of an annual isotopic cycle because our sampling methodology mixed snow from only a 5–10 cm thick layer at 1 m depth. If a seasonal maximum or minimum happened to fall at 1 m depth, the resulting $\delta^{15}N_{NO3}$, $\delta^{18}O_{NO3}$, and $\Delta^{17}O_{NO3}$ values could be offset from the true annual mean value by 20–50 ‰, 10–20 ‰, and 5–6 ‰, respectively (Figure 4).

For example, although the oxygen isotopic values for 1 m depth samples at CHIC-18 and CHIC-20 are much higher than expected (see high values near 130 kg m$^{-2}$ a$^{-1}$ in Figure 5), their values are similar to winter maximum values and may simply be a result of seasonally-biased sampling. Future sampling of 1 m depth samples should ideally mix snow from at least a 50 cm range (i.e., from 1.0 to 1.5 m depth) to reduce the chance of seasonal bias and provide more accurate $\omega(NO_3^-)$ and NO$_3^-$ isotopic values, and ideally the exact mixing depth could be adapted in advance for any site based on modeled

accumulation and compaction rates.

The skin layer samples also show an increase in $\delta^{15}N_{NO3}$ with lower SMB (Figure 5b–c) despite not having much photolytic mass loss that would drive this pattern. This spatial relationship between $\delta^{15}N_{NO3}$ and SMB in the skin layer likely results from $NO_3^-$ recycling (Erbland et al., 2015; Winton et al., 2020), where some of the $NO_3^-$ deposited on the skin layer is derived from re-oxidized photolytic $NO_x$ ventilated from the local snowpack. Because $NO_3^-$ in the snowpack beneath the skin layer has higher $\delta^{15}N_{NO3}$ at drier sites due to increased photolytic mass loss, the isotopic ratios of photolyzed $NO_x$ products and resulting re-oxidized $NO_3^-$ coming from the snowpack will also tend to have higher $\delta^{15}N_{NO3}$ values at drier sites. Additionally, the skin layer $NO_3^-$ would sit at the surface for a slightly longer period at the drier sites than the wetter sites, potentially also giving a slightly greater photolytic imprint on skin layer $\delta^{15}N_{NO3}$ for sites with lower SMB.

This shared spatial relationship with SMB for both the skin layer and 1 m depth $\delta^{15}N_{NO3}$ samples might be seen as evidence that the $\delta^{15}N_{NO3}$ values at 1 m are simply preserving an already existing spatial relationship in $NO_3^-$ isotopes present in the skin layer. If photolytic impacts were indeed the same at all sites, regardless of SMB, we would expect the slope of the 1 m depth samples to match the slope of the skin layer samples, because the degree of isotopic fractionation per unit depth would be the same at every site. However, comparing the spatial regressions of the skin layer samples to the 1 m depth samples reveals that both skin layer and 1 m depth samples have higher $\delta^{15}N_{NO3}$ values as SMB decreases, but the $\delta^{15}N_{NO3}$ values in the 1 m samples increase at a greater rate than the skin layer samples (i.e., the magnitude of the regression's slope is greater for the 1 m depth dataset than for the skin layer dataset) (Table 2). As a result, the greater photolytic action at drier sites enhances and exaggerates the pre-existing $\delta^{15}N_{NO3}$ trend with SMB observed in the skin layer.

The oxygen isotope regressions with SMB also provide some evidence of greater photolytic activity at drier sites. Making definitive conclusions from the oxygen isotope regressions is more difficult than for nitrogen isotopes because the uncertainties of the 1 m depth layer regressions largely overlap and encompass the regressions for skin layer samples. Still, the regressions suggest that at sites with the highest SMB (180–200 kg m$^{-2}$ a$^{-1}$), there will not be a significant difference in the oxygen isotopic ratios between the skin layer and the 1 m depth layer while, in contrast, the skin layer has higher $\delta^{18}O_{NO3}$ and $\Delta^{17}O_{NO3}$ values than 1 m depth at the driest SMB sites (110–130 kg m$^{-2}$ a$^{-1}$). This divergence is also expressed through the regression slopes where the 1 m depth samples have a more positive relationship with SMB than the skin layer samples (Figure 4c–d).

Compared to nitrogen isotopes, it thus appears that a greater degree of photolytic activity (i.e., a drier site) is needed to observe a clear divergence between skin layer and 1 m samples for oxygen isotopic values. This is a reasonable observation because the apparent isotopic fractionation factors for oxygen isotopes are much smaller than for nitrogen, and we would expect based on these observations that photolytic impacts become obvious more quickly for $\delta^{15}N_{NO3}$ than for $\delta^{18}O_{NO3}$ or $\Delta^{17}O_{NO3}$. However, the reduced photolytic impact in oxygen isotopes compared to nitrogen isotopes may seem surprising given that the isotopic trends with depth in the pit data are much clearer in the oxygen isotopes than $\delta^{15}N_{NO3}$ (Figure 4).

The relatively limited photolytic signal in the oxygen isotopes is likely due in part to summer bias in our skin layer sampling whereas the 1 m depth samples draw from the full range of the annual cycle. During summer, skin layer $NO_3^-$ has maximum $\omega(NO_3^-)$ and minimum isotopic values (Figure 1). An annual mean skin layer sample, however, would have lower $\omega(NO_3^-)$ and higher isotopic values, although still weighted heavily toward summer values due to summer's much higher $NO_3^-$ concentrations. Adjusting our observed skin layer values to reflect annual values increases our calculated $\varepsilon_{app}$ values for all isotopic species by 3–10 ‰. This slightly weakens the observed $^{15}\varepsilon_{app}$ values for nitrogen but more impactfully shifts the $^{18}\varepsilon_{app}$ and $^{17}\varepsilon_{app}$ of the oxygen isotopes to clearly positive values that better reflect our observations of oxygen isotopic change in the pit profiles.

The snow pit isotopic trends reveal another unusual characteristic that may also help explain why skin layer and 1 m depth sample values only diverge at drier sites for oxygen isotopes. In the snow pits, $\delta^{15}N_{NO3}$ values rapidly increase in the

uppermost 5–10 cm coinciding with the rapid decline in $\omega(NO_3^-)$ (Figure 4a–b). This follows our expectations as photolytic activity is concentrated near the surface due to rapid attenuation of solar radiation in the snowpack. However, $\delta^{18}O_{NO3}$ and $\Delta^{17}O_{NO3}$ values exhibit a steady decline throughout the entire 100–200 cm depth of the pits with no obvious signs of a greater
rate of decline at shallow depths where photolytic activity should be strongest (Figure 4c–d). Re-examining the drivers of $NO_3^-$ oxygen isotopic change may help explain this inconsistency.

The seeming insensitivity of oxygen isotopic values to changing photolytic activity may instead reflect changes in the balance between two competing isotopic effects. Although it has not been experimentally observed, photolytic mass loss is theoretically predicted to have a direct isotopic fractionation effect on oxygen that would increase $\delta^{18}O_{NO3}$ values in the
remaining $NO_3^-$, similar to $\delta^{15}N_{NO3}$ (Frey et al., 2009). This is counter-balanced by an opposing isotopic effect resulting from oxygen exchange from a cage effect (McCabe et al., 2005). In uppermost 5–10 cm of the snowpack, the proximity of the atmosphere makes it relatively easy for photolyzed $NO_3^-$ to be lost from the snowpack. This leads to the rapid change observed in $\delta^{15}N_{NO3}$, but the lack of corresponding substantial change in $\delta^{18}O_{NO3}$ suggests that the isotopic effect of mass loss fractionation is balanced by the competing effect from oxygen exchange in these uppermost depths.

Deeper within the photic zone, however, photolyzed $NO_3^-$ lacks this nearby interface with the atmosphere, and it is more likely that photolytic products will re-oxidize back into $NO_3^-$ in place or somewhere within the photic zone. This increase in intra-snowpack $NO_3^-$ recycling will reduce photolytic mass loss fractionation, but oxygen exchange can still occur. The balance in competing isotopic effects will thus shift increasingly toward oxygen exchange with greater depth until photolytic activity ceases due to complete light attenuation. Although photolytic activity and $NO_3^-$ re-oxidation is decreasing with
depth due to radiation attenuation, the increased dominance of the oxygen exchange effect appears to compensate for the decreasing radiation to produce the steady lowering in $\delta^{18}O_{NO3}$ values. Therefore unlike $\delta^{15}N_{NO3}$, the greatest degree of isotopic change for $\delta^{18}O_{NO3}$ should occur beneath the immediate uppermost snowpack layers once the oxygen exchange effect is predominant. Presumably, the quicker burial of $NO_3^-$ at wetter sites would limit the amount of oxygen exchange that could occur in the deeper photic zone, and thus we observe little difference in $\delta^{18}O$ values between skin layer and 1 m depth
samples. In contrast, the greater photolytic activity at drier sites would enhance the imbalance between competing isotopic effects and produce distinctly lower $\delta^{18}O_{NO3}$ values at 1 m depth compared to the surface.

This proposed concept works well to explain the patterns observed in $\delta^{18}O_{NO3}$, but it struggles to fully explain the similar patterns also observed in $\Delta^{17}O_{NO3}$. Unlike $\delta^{18}O_{NO3}$, photolytic mass loss is not expected to affect $\Delta^{17}O_{NO3}$ values (McCabe et al., 2005). Thus, there is no isotopic effect counter-balancing the oxygen exchange brought by $NO_3^-$ re-oxidation and the
cage effect, yet we still observe an unusually steady lowering of $\Delta^{17}O_{NO3}$ values with depth in the pit data. As previously mentioned, periods with high photolytic mass loss observed in the pit data (as indicated by the highest $\delta^{15}N_{NO3}$ values) often have $\Delta^{17}O_{NO3}$ peaks that are higher than would be expected compared to the coinciding $\delta^{18}O_{NO3}$ values. In other words, $\Delta^{17}O_{NO3}$ values decline from skin layer values to a lesser extent than $\delta^{18}O_{NO3}$ values during times of high photolytic mass loss, which is in fact the opposite expected from our proposed "balanced competing effects" concept and difficult to explain
mechanistically. Additionally, none of these ideas can explain why $\delta^{18}O_{NO3}$ and $\Delta^{17}O_{NO3}$ values appear to keep declining well beneath the lower photic zone limit in P5. Overall, this suggests that substantial complexities and unknowns still exist with regards to photic zone processes and $NO_3^-$ dynamics at the snow-atmosphere interface in Antarctica and resolving these issues will be necessary to properly interpret $NO_3^-$ oxygen isotopes archived in Antarctic ice.

## 6. Conclusions
Our analysis of $NO_3^-$ in snow samples taken along the CHICTABA transect reveals the environmental drivers of $NO_3^-$ concentration and isotopic variability at an unprecedented spatial resolution for a region of East Antarctica with intermediate

SMB values (110–200 kg m$^{-2}$ a$^{-1}$). We find that seasonal geochemical cycles observed in atmospheric NO$_3^-$ are preserved in NO$_3^-$ buried in the snowpack. However, these cycles are clearly altered by post-depositional changes as shown by NO$_3^-$ isotopic values and calculated apparent isotopic fractionation factors that match observations from elsewhere in Antarctica attributed to photolysis. Additionally, we observe that the isotopic changes are greater at drier sites along the transect. This is consistent with photolysis as a causative factor in NO$_3^-$ isotopic change because slower burial rates at dry sites expose NO$_3^-$ to more cumulative photolytic radiation before the NO$_3^-$ is buried beneath the reach of sunlight.

Because photolysis does not entirely wipe out the initial seasonal NO$_3^-$ cycles like it does at very dry sites in the Antarctic interior (e.g., Erbland et al., 2013; Shi et al., 2015), the interpretation of NO$_3^-$ is complicated in firn and ice cores from regions with intermediate SMB values. If sampled at a high enough resolution, seasonal cycles in NO$_3^-$ concentration and isotopes may be recoverable far into the past, but these values are not representative of the exact NO$_3^-$ character at the time of deposition. Photolysis will reduce $\omega$(NO$_3^-$) while increasing $\delta^{15}$N$_{NO3}$ values and decreasing $\delta^{18}$O$_{NO3}$ and $\Delta^{17}$O$_{NO3}$ values from their initial atmospheric values. The degree of photolytic change is not likely consistent from year to year as it will depend strongly upon local SMB. Because regions in East Antarctica with intermediate SMB are generally found on the sloped transition between the high elevation interior plateau and low-lying coastal zone, katabatic winds drive intense irregular erosion and deposition of the snow surface (Frezzotti et al., 2002; Agosta et al., 2012). Additionally, intrusions by atmospheric rivers and lower latitude moisture bring infrequent but regular extreme accumulation events to these transitional regions (Gorodetskaya et al., 2014; Wille et al., 2021; Djoumna and Holland, 2021). As a result, the regions have very high interannual SMB variability that leads to very high interannual variability in photolytic impacts that makes it difficult or impossible to reconstruct precise initial atmospheric NO$_3^-$ characteristics at a seasonal resolution from NO$_3^-$ archived in firn and glacial ice.

However, relative to the interannual variability introduced by local SMB changes, interannual differences in mean atmospheric NO$_3^-$ isotopic values are likely to be relatively small, at least in the recent past. Atmospheric and skin layer NO$_3^-$ samples at Dome C are generally consistent year to year (Erbland et al., 2013; Winton et al., 2020), and atmospheric NO$_3^-$ observed at other sites have similar patterns and values (Wagenbach et al., 1998; Savarino et al., 2007; Frey et al., 2009). Regular sampling of atmospheric and skin layer NO$_3^-$ over one or more full years at an intermediate SMB site would greatly aid our comprehensive spatial understanding of NO$_3^-$ depositional dynamics, but unfortunately no permanent scientific stations exist in intermediate SMB regions far from the coast. The most practical approach to NO$_3^-$ interpretation in firn and ice cores from intermediate SMB sites may be to assume atmospheric NO$_3^-$ isotopic values can be considered "constant" when aggregated over multiple years. As a result, observed isotopic variability at this multiannual resolution will reflect changes in photolytic activity driven by SMB, with stronger and more detectable effects at drier sites and more accuracy with more years of accumulation aggregated per sample.

Recognizing the importance of SMB in determining the isotopic composition of NO$_3^-$ may allow us to investigate other drivers of isotopic change. Ice cores from intermediate accumulation regions can preserve seasonal ion and water isotope cycles well enough to produce highly precise chronologies (Buizert et al., 2015). Coupled with physical measurements of the ice core's volume and mass, we can model SMB based on physical changes in ice density and/or annual layer thickness (e.g., Fudge et al., 2016; Akers et al., 2022). This physical SMB reconstruction could then be used to remove the SMB signal from a parallel NO$_3^-$ isotope record, and the residual NO$_3^-$ isotopic variability should reflect past changes in other environmental factors, such as insolation, total column ozone, snow optical properties, and atmospheric NO$_3^-$ sourcing and chemistry (Zatko et al., 2016; Cao et al., 2022; Shi et al., 2022b). This would be most effective for $\delta^{15}$N$_{NO3}$ which has a more clear relationship with SMB (Akers et al., 2022) than $\delta^{18}$O$_{NO3}$ or $\Delta^{17}$O$_{NO3}$, but additional investigation into the mechanisms behind the apparent impacts of photolysis on oxygen isotopic composition is likely to provide valuable insight into past and present NO$_3^-$ dynamics as well. Additionally, ice cores taken from high SMB regions nearer the coast (i.e., regions with limited

photolytic mass loss such as Law Dome) should better preserve the seasonal and interannual variability of atmospheric $NO_3^-$ and can provide an interesting comparison for ice core $NO_3^-$ records from drier inland settings.

Our $NO_3^-$ work as part of CHICTABA adds to the growing body of literature on $NO_3^-$ isotopes that point the way forward for future improvements to $NO_3^-$ interpretation in Antarctica. This knowledge is particularly critical for understanding the environmental changes archived in deep Antarctic ice cores, including new projects such as Beyond EPICA-Oldest Ice (Lilien et al., 2021). Based on our CHICTABA findings and other recent studies (Erbland et al., 2013; Shi et al., 2015, 2018a), we highlight in particular the value of $NO_3^-$ isotopic profiles from snow pits in understanding the transition of $NO_3^-$ from the atmosphere into archived glacial ice. We argue for additional dedicated pit sampling of $NO_3^-$ isotopes with particular emphasis on extending profile depth below 1 m with paired chronological and snow density profiles to constrain SMB changes. Replication of pit profiles at individual sites will also improve our understanding of the natural range of local spatial $NO_3^-$ variability. Expansion of atmospheric $NO_3^-$ monitoring beyond Dome C and Zhongshan stations will also help constrain spatial variability in seasonal $NO_3^-$ cycling. Finally, the potential spatial variability in snow optical properties and photic zone depths remain one of the greatest unknowns in Antarctic $NO_3^-$ dynamics (France et al., 2011, 2020; Winton et al., 2020), and improved field observations and modeling will be required to precisely interpret $NO_3^-$ isotopic variability for paleoenvironmental reconstructions. Overall, the $NO_3^-$ samples from the CHICTABA mission confirm the general understanding of $NO_3^-$ dynamics in East Antarctica that has developed in the past two decades and suggest that the understudied regions between the coasts and interior dome summits hold much untapped potential to improve our understanding of the Antarctic environment.

**Acknowledgements**

We express thanks to the following individuals for project assistance and data support: Sarah Albertin, Albane Barbero, Mathieu Casado, Armelle Crouzet, Patricia Martinerie, Jean Martins, Vincent Favier, Elsa Gautier, Alexis Lamothe, and the overwintering crews at Concordia station. We particularly recognize Cécile Agosta, Charles Amory, and Christophe Kittel for their supply of MARv3.12.1 data and assistance. We acknowledge the logistical support of IPEV for the French missions in Antarctica, the IPEV and PNRA colleagues and overwintering crews at Concordia station, the CHICTABA traverse team for fieldwork assistance, and the Air-O-Sol facility at IGE for microbial culturing. We thank our two reviewers for their feedback and helpful suggestions.

**Data availability**

Data are available through the PANGAEA online repository at https://doi.org/10.1594/PANGAEA.948355.

**Code availability**

All code used to analyze data and produce figures is available at https://doi.org/10.5281/zenodo.7287413.

**Funding**

European Commission European Horizon Marie Skłodowska-Curie individual fellowship 889508: SCADI;

Institut National des Sciences de l'Univers-CNRS: LEFE-IMAGO project PROXYNNOV;

Agence Nationale de la Recherche (ANR): ANR-10-LABX56, ANR-11-EQPX-009-CLIMCOR, ANR-16-CE01-0011-01-EAIIST;

French Polar Institute IPEV: programs 1115 (CHICTABA), 1117 (CAPOXI 35-75), and 1169 (EAIIST).

## Author contributions

Conceptualization: PDA, JS, NC

Investigation: PDA, JS, NC, OM, ELM

Formal analysis: PDA

Visualization: PDA

Funding acquisition: PDA, JS

Writing – original draft: PDA

Writing – review & editing: PDA, JS, NC, OM, ELM

The authors declare that they have no competing interests.

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
