# Peer review of "Photolytic modification of seasonal nitrate isotope cycles in East Antarctica"

_EGUsphere, 2022_

## Referee Comment (RC2)

Comments on Akers et al. "Photolytic modification of seasonal nitrate isotope cycles in East Antarctica"

Akers et al. reported new dataset of nitrate concentrations and isotopic composition in snow collected along traverse with generally intermediate snow accumulation rate (100-200 kg m$^{-2}$ a$^{-1}$) in East Antarctica. The authors collected three types of snow samples, the uppermost skin layer, the 100cm depth snow, and snow pits samples, and then they analyzed the concentrations, and $\delta^{15}$N, $\delta^{18}$O and $\Delta^{17}$O of nitrate in these samples. With these data, the authors made an assessment of the degree of nitrate photolysis among different sites. They concluded that the primary deposition signals may be largely preserved, especially in the high snow accumulation rate sites, but the photolytic loss of nitrate occurs, with a higher degree of loss in drier sites. These findings are of significance to interpret the nitrate records in ice cores recovered from the East Antarctica where a large part of regions feature the intermediate snow accumulation rate. In general, this paper is well written/organized and fits the scope of Atmospheric Chemistry and Physics. Congratulations to the authors for that they collected so many snow samples in the very harsh Antarctic environments and provided the comprehensive nitrate isotope dataset. I recommend the publication of the work, and I only have several comments that may help improve/clarify the interpretation of the data.

First, the authors made an extensive interpretation of the 100cm depth data. The 100cm depth snow samples were collected by mixing a 5-10cm thick layer of snow from 100cm below the surface. Considering that snow accumulation rate ranges from ~100 to 200 kg m$^{-2}$ a$^{-1}$ along the traverse in this study, this 5-10 cm snow layer likely only covers a single season. In this case, it is unknown what the 100cm depth snow can present (summertime or wintertime snow, or a mixing?). It seems unlikely to date the 100cm snow without a snow pit sampling, considering that the snow accumulation varies both spatially and temporally. Based on this, I donot think it is reasonably to calculate/interpret the apparent fractionation constant ($\varepsilon_{app}$) using the skin layer data versus the 100cm snow data (note that you only have two points for $\varepsilon_{app}$ calculation) at one site. At a specific site, the 100cm snow from wintertime or summertime deposition may result in distinct $\varepsilon_{app}$ values. From Table 3, the $^{15}\varepsilon_{app}$ varies between about -20 and -70‰, and both positive and negative $^{18}\varepsilon_{app}$ and $^{17}E_{app}$ are present along the traverse. It is suspected that the sampled 100cm snow at different sites may be from different season deposition. Then, comparison between the skin layer and 100cm snow data will be challenging, because the 100cm snow may from different seasons at different sites. In my opinion, the interpretation of the 100cm snow data would be uncertain before clarifying what the sample/data can present, though I noticed that the authors have clearly pointed out the uncertainty of the 100cm snow at the end of the discussion.

Second, the authors dated the snow pits based on the concentrations and isotopic composition of nitrate. Do they have other parameters such as water isotopes and other chemical ions that were usually used for snow/ice dating? In addition, the authors are encouraged to clarify how they separate the snow pit sample groups (warm season versus cold season). In my opinion, the snow pit data are rather valuable as these pits cover different snow accumulations, and thus the snow pit data may be explored further. For instance, is it possible to quantify the overall degree of snow nitrate photolysis degree at different sites and its relation to snow accumulation rate? From Figure 4, it seems that photolytic processing at P1 is rather minor? Or based on these comprehensive snow pit data, is it possible to suggest a threshold of snow accumulation (or an accumulation range), above which the photolytic processing of nitrate

would be minor? Because previous investigations based on nitrate concentration data have proposed that post-depositional loss become insignificant at sites with snow accumulation rate above ~100kg m$^{-2}$ a$^{-1}$. This would be of great significance to the precise interpretation of ice core nitrate (and its isotopes) records in the regions with intermediate sow accumulation rate.

Other minor/specific comments:

The Introduction section, in my opinion, is generally good.

L36-40, please also consider a recent work on both atmospheric and snowfall nitrate annual cycles at a coastal site, East Antarctica. (Shi, G., Li, C., Li, Y., Chen, Z., Ding, M., Ma, H., Jiang, S., An, C., Guo, J., Sun, B., Hastings, M.G., 2022. Isotopic constraints on sources, production, and phase partitioning for nitrate in the atmosphere and snowfall in coastal East Antarctica. Earth and Planetary Science Letters 578, 117300.)

L107, could you please be specific about the unknowns regarding atmospheric nitrate budget in Antarctica?

L119-121, Erbland et al. (2013) suggested an overall oxygen isotopic equilibrium at the Dome C air-snow interface?

Figure 1, the unit of mass fraction of nitrate in the atmosphere should be ng m$^{-3}$, not ng g$^{-1}$.

Section2.3, maybe several factors such as stratospheric ozone (TCO) influencing snow nitrate photolysis can be included, in addition to snow accumulation rate.

L156, a space is needed between $NO_3^-$ and cycles

L160-162, a good point here. In addition to the downward transport of NOx may undergo oxidation to form nitrate, NOx from snow nitrate photolysis in the interstatial air may also be re-oxidized before it is pumped by wind to the overlying atmosphere. Now, we know nothing about this process.

L189-193, both nitrate concentration measurements and snow sample pre-concentration were conducted at Dome C?

L201, what do you mean by "$\pm 0.7–1.1$" and the other values?

L270-271, how did you remove the linear trends of the data?

L317-318, I donot the apparent fractionation constants are generally consistent at different sites according to the following description: $^{15}\varepsilon_{app}$ varies between -25 and -66 ‰, and both positive and negative values are for the $^{18}\varepsilon_{app}$ and $^{17}E_{app}$. In my opinion, the negative $^{18}\varepsilon_{app}$ and $^{17}E_{app}$ values may be associated with that the sampled 100cm depth snow is the wintertime deposition which features elevated oxygen isotopic ratios of nitrate. Even after 2-3 years of photolysis, these values may be still higher than the skin layer values (summertime deposition). Again, this point raises the question that what the depth snow sample can present.

L354, DDU and Zhongshan are the only available observations on the complete isotopes of atmospheric nitrate in coastal Antarctica.

L368-370, the apparent fractionation of oxygen isotopes is related to both the photolytic loss and re-oxidation of photoproducts in the snow grain condensed phase. These two processes dominate the $^{18}\varepsilon_{app}$ and $^{17}E_{app}$. Up to date, the oxygen isotopic effects of a single photolytic loss process remain unknown.

L410-415, if the very clear negative trends in oxygen isotope values with depth in P2 and P3 suggest the cumulative effects of snow nitrate photolysis, the synchronously increase trends in $\delta^{15}N$ would be also expected assuming the relatively constant atmospheric inputs of nitrate. But the observations show no clear trends in $\delta^{15}N$. Can the authors explain this difference?

L447-454, is it possible that the "skin layer" nitrate has experienced some extent of photolytic processing, though the extent may be rather small. Considering that the sample skin layer thickness is comparable at different sites, and the varied snow accumulation rates would result in the different exposure time of the "skin layer" at the sampling sites. Then, the higher $\delta^{15}N$ and lower oxygen isotopic ratios would be expected at the low snow accumulation sites. Is this maybe an additional reason for the observed relationships?

L 473-475, from Figure 4, it seems that the cumulative effects of photolytic loss are more evident on oxygen isotopes than on the nitrogen isotope (the trends in oxygen isotopes are more significant)?

**Supplementary information:**
L50-55, caption of Figure S2, typos, please check.

---

## Author Response (AR1)

**Review Response**

We thank both our reviewers for their time devoted to reading our manuscript and suggesting improvements. Our answers to their comments are listed here.

**Reviewer 1**

**Based on this, I do not think it is reasonably to calculate/interpret the apparent fractionation constant (εapp) using the skin layer data versus the 100cm snow data (note that you only have two points for εapp calculation) at one site. At a specific site, the 100cm snow from wintertime or summertime deposition may result in distinct εapp values. From Table 3, the 15εapp varies between about -20 and -70‰, and both positive and negative 18εapp and 17Eapp are present along the traverse. It is suspected that the sampled 100cm snow at different sites may be from different season deposition. Then, comparison between the skin layer and 100cm snow data will be challenging, because the 100cm snow may from different seasons at different sites. In my opinion, the interpretation of the 100cm snow data would be uncertain before clarifying what the sample/data can present, though I noticed that the authors have clearly pointed out the uncertainty of the 100cm snow at the end of the discussion.**

We added this recognition of the limits of our 1 m depth samples into the methods and earlier into the results and discussion to make sure the reader could better put the data into context in terms of accuracy and representativeness. This point is now mentioned multiple times throughout the manuscript.

**Do they have other parameters such as water isotopes and other chemical ions that were usually used for snow/ice dating? In addition, the authors are encouraged to clarify how they separate the snow pit sample groups (warm season versus cold season).**

Unfortunately, the only data that were collected on the pit snow samples were nitrate concentration and nitrate isotopic values. We have added these lines into the results to clarify how we identified seasonal cycles:

In the absence of other supplemental geochemical data, the residuals of the $\Delta^{17}O_{NO3}$ regression with depth were used to identify seasonal cycles with positive residuals representing colder months and negative residuals representing warmer months (**Error! Reference source not found.**). This seasonal identification is based on Dome C $NO_3^-$ monitoring data (**Error! Reference source not found.**) and previously reported seasonal $\Delta^{17}O_{NO3}$ cycles linked to snow $\delta^{18}O$ variability in a snow pit (Shi et al., 2015).

**In my opinion, the snow pit data are rather valuable as these pits cover different snow accumulations, and thus the snow pit data may be explored further. For instance, is it possible to quantify the overall degree of snow nitrate photolysis degree at different sites and its relation to snow accumulation rate? From Figure 4, it seems that photolytic processing at P1 is rather minor? Or based on these comprehensive snow pit data, is it possible to suggest a threshold of snow accumulation (or an accumulation range), above which the photolytic processing of nitrate would be minor? Because previous investigations based on nitrate concentration data have proposed that post-depositional loss become insignificant at sites with snow accumulation rate above ~100kg m-2 a-1. This would be of great significance to the precise interpretation of ice core nitrate (and its isotopes) records in the regions with intermediate sow accumulation rate.**

We agree that this proposed line of investigation would provide very valuable insight into nitrate dynamics. Unfortunately for the pits in this study, we don't have snow density measurements

outside of a few shallow cores that are not necessarily representative of the hyperlocal density profiles of the pits themselves. Combined with the lack of other geochemical data that could be used to create a potentially better chronology than our estimates using $\Delta^{17}O_{NO3}$, any attempt to create a pit specific profile of annual accumulations would probably be so uncertain as to be a mere guess. Regarding P1, it is unclear whether there truly is not much photolytic loss at this site in generally or if this pit simply was sampled in a spot with a locally higher accumulation due to a drift. The $\delta^{15}N_{NO3}$ and $\delta^{18}O$ values at P1 at their most extreme would appear to reflect some level of photolytic activity, although we note that their values ($\sim\delta^{15}N \approx 50$ ‰, $\delta^{18}O < 60$ ‰) are within values observed in Dome C skin layer snow, although pretty much equal to the most extreme values observed at Dome C. We have added more discussion about the pit photolysis through comparison between nitrogen and oxygen isotopes.

Overall, this highlights the need for more focus on snow pit sampling of $NO_3^-$, with particular focus on getting multiple profiles per site (or better spatial aggregation per site) and having paired chronological and density profiles. We've added this point to our conclusions.

**Please also consider a recent work on both atmospheric and snowfall nitrate annual cycles at a coastal site, East Antarctica. ( Shi, G., Li, C., Li, Y., Chen, Z., Ding, M., Ma, H., Jiang, S., An, C., Guo, J., Sun, B., Hastings, M.G., 2022. Isotopic constraints on sources, production, and phase partitioning for nitrate in the atmosphere and snowfall in coastal East Antarctica. Earth and Planetary Science Letters 578, 117300.**

Thanks for the suggestion, it has now been added as a citation and as information at multiple points.

**L107, could you please be specific about the unknowns regarding atmospheric nitrate budget in Antarctica?**

We made a rewording and clarification of the sentence to highlight that we are referring to discrepancies between field and model values, as discussed more in the cited papers.

**L119 121, Erbland et al. (2013) suggested an overall oxygen isotopic equilibrium at the Dome C air snow interface?**

Erbland 2013 attributed the difference between atmospheric and snow surface oxygen values as being a lagged effect from the difference in reservoir sizes between the atmosphere and snow surface. However, they were working with only a couple of years of data and the consistent offset, particularly in the early winter, wasn't as clear (versus it being an effect of a temporal shift or lag). The reservoir equilibrium theory also struggles to explain why the snow surface reservoir has higher oxygen isotopic values than any observations of atmospheric $NO_3^-$ (that is to say, how is the larger reservoir getting to these high of $\delta^{18}O$ and $\Delta^{17}O$ values if the atmospheric input never has isotopic values that high and we don't have a clear fractionation process described that would do this).

**Figure 1, the unit of mass fraction of nitrate in the atmosphere should be ng m 3 , not ng g 1**

This has now been corrected in the figure and in the caption.

**2.3, maybe several factors such as stratospheric ozone TCO influencing snow nitrate photolysis can be included, in addition to snow accumulation rate**

We added a line to acknowledge these other possible factors and discuss them more in the discussion.

**L156, a space is needed between NO 3 and cycles**

Added.

**L160-162, a good point here. In addition to the downward transport of NOx may undergo oxidation to form nitrate, NOx from snow nitrate photolysis in the interstatial air may also be re oxidized before it is pumped by wind to the overlying atmosphere. Now, we know nothing about this process.**

Yes, we certainly see some evidence of this process, but it is hard to directly observe. We add some discussion of this concept in our oxygen isotope discussion.

**L189-193, both nitrate concentration measurements and snow sample pre concentration were conducted at Dome C?**

Yes, and we added a couple of words to clarify this all happened quickly after melting.

**L201, what do you mean by 0.7 1.1 and the other values?**

These are the root mean square errors of the calibrated standards to give a sense of the uncertainty of our analytical process. We've reworded this sentence to be more clear.

**L270-271, how did you remove the linear trends of the data?**

We correlated the residuals of the linear regressions, which presents the data as anomalies around the linear trend. This in effect removes the linear trend for correlation.

**L317-318, I do not the apparent fractionation constants are generally consistent at different sites according to the following description : 1515εapp varies between -25 and -66 ‰, and both positive and negative values are for the 18εapp and 17Eapp. In my opinion, the negative 18εapp and 17Eapp values may be associated with that the sampled 100cm depth snow is the wintertime deposition which features elevated oxygen isotopic ratios of nitrate. Even after 2-3 years of photolysis, these values may be still higher than the skin layer values (summertime deposition). Again, this point raises the question that what the depth snow sample can present.**

Yes, the "generally consistent" phrasing was poor choice. We've changed the phrasing throughout to make it clear we acknowledge the wide variability and discuss its likely source in the sampling methodology.

**L354, DDU and Zhongshan are the only available observations on the complete isotopes of atmospheric nitrate in coastal Antarctica.**

Shi 2022 cited here and discussed a bit now.

**L368-370, the apparent fractionation of oxygen isotopes is related to both the photolytic loss and re oxidation of photoproducts in the snow grain condensed phase. These two processes dominate the 18εapp and 17Eapp. Up to date, the oxygen isotopic effects of a single photolytic loss process remain unknown.**

We have rephrased this paragraph and others to make it more clear that the apparent fractionation factors for the oxygen species are consistent to what has been observed in oxygen isotopes at other photolysis-dominated sites (incorporating both photolysis and the recycling effects), but these apparent fractionation factors are not a direct photolytic fractionation factor.

**L410-415, if the very clear negative trends in oxygen isotope values with depth in P2 and P3 suggest the cumulative effects of snow nitrate photolysis, the synchronously increase trends in δ**

**15 N would be also expected assuming the relatively constant atmospheric inputs of nitrate. But the observations show no clear trends in δ 15 N . Can the authors explain this difference**

We have added discussion regarding this clear discrepancy. We now give greater focus to the effect of oxygen atomic exchange and highlight how it might be influencing oxygen isotopic ratios differently than nitrogen. Namely, we stop using the decline in oxygen isotopic values as a "sign of photolysis" and discuss it in terms of oxygen atomic exchange which is triggered by photolysis.

**L447-454, is it possible that the skin layer nitrate has experienced some extent of photolytic processing, though the extent may be rather small. Considering that the sample skin layer thickness is comparable at different sites, and the varied snow accumulation rates would result in the different exposure time of the skin layer at the sampling sites. Then, the higher δ 15 N and lower oxygen isotopic ratios would be expected at the low snow accumulation sites . Is this maybe an additional reason for the observed relationships?**

This is a good hypothesis, and we've added it into the discussion. Thanks!

**L 473 -475, from Figure 4, it seems that the cumulative effects of photolytic loss are more evident on oxygen isotopes than on the nitrogen isotope (the trends in oxygen isotopes are more significant)?**

We've added discussion now addressing this issue. We point out why we think the trends in oxygen are so clear when their fractionation values are so weak. Possible explanations include: 1) the skin layer $\delta^{15}N$ values are always much lower than other pit values, whereas the oxygen isotope skin layer values overlap with much of the overall pit range, 2) differences in where photolytic mass loss peaks versus the relative strength of oxygen atomic exchange, and 3) a summer bias to the skin layer values which makes the $\varepsilon_{app}$ for oxygen isotopes appear weaker than it actually is.

**Supplementary information:**

**L50-55, caption of Figure S2 , typos , please check.**

Fixed typos and sub/superscripts.

**Reviewer 2:**

**It seems like a missed opportunity to not compare this work with expectations predicted with the modeling work of Zatko, Alexander, and others (https://acp.copernicus.org/articles/16/2819/2016/). It's important to give the field in general more tools for exploring larger scale implications/interpretation of these types of records. and the Zatko modeling work had so little data to compare with in modeling the entire Antarctic interior.**

We have added references to Zatko et al., 2016 throughout the manuscript and in particular to the discussion of relationship between nitrate recycling and photolytic loss. We do recognize the great value and work included in that paper, and did not intend to omit it from being referenced in this work. However, we note that most of the data in that paper are presented in a series of small maps, and to our knowledge, there is no publicly accessible way to examine its model outputs at a scale more relevant to our work here (e.g., we struggled to do a more in depth comparison when we are largely looking at plotted maps

in the article and trying to eyeball where our transect fits on them). That said, we did try to include the article and its findings where relevant.

**My only other general comment is that it would be worth some extra attention being paid to suggestions for an independent proxy of photolysis. As the literature stands now, several works in Antarctica have shown evidence that d15N of nitrate IS a proxy for column ozone/extent of photolysis. Given the other published work by these same authors putting forward d15N as a proxy for SMB it seems worthwhile to think through more carefully what the criteria is for another proxy that would make d15N even more useful and under what SMB conditions is d15N already useful (ie we would not need this independent proxy).**

We have added a paragraph to the conclusions that discusses how variables such as insolation, TCO, and snow optical properties could be examined in better detail if the SMB effect could be removed by comparison with a physical or other SMB record. We also add a line in the introduction pointing out that these other influences exist and should be detectable under certain conditions. We also highlight that oxygen isotopes are still relatively poorly understood and may hold valuable information regarding these other interpretations of $NO_3^-$ isotopes.

**In the results (~ line 275) it is reported that there is a lack of correlation between d15N and d18O, but there is a significant correlation with D17O. Yet, d18O and D17O are well correlated (r=0.51 "fairly strong positive correlation"). What impact does the larger analytical error for d18O measurements have on these relationships? I recognize the authors link this instead to an offset in the timing of the d18O cycle compared to d15N, but this may be something that should be a bit more highlighted in the discussion. For instance can the offset be quantified? Clearly the d18O can be directly impacted by the loss of nitrate while the D17O would remain the same, so what does the other 49% of varability in d18O and D17O teach us? Are the slopes amongst the different sites different in a way that can be explained physically? While I appreciate the reporting of these relationships, they are not necessarily instructive of any particular mechanistic understanding as the manuscript is written currently.**

The uncertainty of the $\delta^{18}O$ values are still small relative to the overall range of variability in the seasonal cycles and do not seem to be a cause of this correlation discrepancy. We have added the following sentence which offers another explanation: Additionally, $\Delta^{17}O_{NO3}$ values tend to peak higher than $\delta^{18}O_{NO3}$ values when coinciding with the highest $\delta^{15}N_{NO3}$ values (e.g., P2: 75 cm, P3: 35 cm, P4: 55 cm), and these shared extreme values promote a stronger correlation.

Throughout the paper, we go into more detail in discussing the drivers of oxygen isotopic changes. We suggest that the observed differences between $\delta^{18}O$ and $\Delta^{17}O_{NO3}$ may be due to the fact that $\delta^{18}O_{NO3}$ should be sensitive to photolytic mass loss while $\Delta^{17}O_{NO3}$ is not, but acknowledge that this is poorly understood due to the competing effects from $NO_3^-$ recycling and cage effects.

**Section 5.1 - it would make sense to include some discussion of the coastal data and interpretation provided by Shi et al (The Cryosphere, 2018) -- www.the-cryosphere.net/12/1177/2018/ .**

Some points from this reference are now cited and included in the manuscript.

**Line 366 -- this should be rephrased. The observed APPARENT fractionation factors are similar to those observed in other FIELD studies but this disagrees with the laboratory and theoretical predictions (especially of Frey et al 2009 cited here). This is all outlined in the introduction of the paper. The apparent fractionations do not necessarily reflect "NO3-photolytic fractionation" since this fractionation(s) are not currently predicted by theory to be positive. This should be rephrased to address that the apparent factors are representative of the impact of NO3- photolytic fractionation (ie there may be competing reaction or re-formation of nitrate in situ in the snow that leads to lowering D17O and d18O as reported in other works already cited within the manuscript).**

We have now adjusted throughout the text to 1) make sure we are stating apparent fractionation factors and 2) that the observed oxygen isotopic changes are not a direct result of photolysis, but rather match what has been observed in other studies where photolysis are occurring and include a component of oxygen atomic exchange.

**Moreover, the skin layer elevated values compared to the snow hint at an indication that maybe the theoretically predicted negative fractionation factors for d18O are being expressed. This should be reflected on a bit more in the discussion here.**

This is now discussed in the section investigating the relationships between isotopes and SMB.

**A schematic representation of the discussion in lines 425-435 would be useful/would make this article more appealiing to a larger readership.**

This figure has been added as Figure 6.

**Line 436 -- again, I take issue with linked a lowering of d18O and D18O specifically to photolysis. It is not the photolysis necessarily that leads to the lowering -- the only mechanisms reported in the literature involve re-oxidation of the photolytic products (ie this cannot be explained by photolysis alone and therefore could vary in different snow environments espeicllay given the large range of apparent factors observed in the field.**

Clarified to state that the lowering is due to reoxidation and atomic exchange, not directly to photolysis.

**Line 470-480 -- the d18O and D17O being "notably higher in the skin layer than at 1 m depth" is not obvious in Figure 5. In fact, given the large uncertainty envelope for the relaionships with SMB it's not a clearly true statement that these values are significantly different. It should be better addressed as to why the skin layers appear to stay relatively consistent across SMB versus at 1m depth. This combined with the higher d15N and**

**higher d18O does seem to hint at direct loss (vs re-combination) happening in that skin layer?**

Rephrased to say that they are generally higher, and that this appears due to the divergence of 1m depth samples becoming lower at drier sites while skin layer samples stay largely constant. Additionally, we have reinterpreted a possible explanation for the difference in slopes. See three new paragraphs 564–594.

**Line 474 -- please phrase as "apparent isotopic fractionation factors"**

Removed in editing, but other sections checked for similar possible issues.

**Line 480 --  please phrase as "part of a annual isotopic cycle" or similar**

Changed as suggested.

**Line 485 -- this is an important recommendation but also needs to be written within the context of also knowing SMB. Is the 50 cm range at 1-1.5 m depth specific to these intermediate SMBs? I think this call to the communtiy should be given more specifciity given different accumulation rates in regions where ice cores are drilled.**

Altered to be more clear about how to adapt this more broadly to any accumulation rate.

**Line 496 -- change "supports" to "is consistent with photolysis". The evidence does not support theoretical predictions of the impact of photolysis on d18O.**

Changed as suggested.

**Line 511 -- is initial NO3- the initially deposited nitrate at the snow surface or is it the "primary" nitrate signal as I have seen it called in other Savarino papers?**

Rephrased as "mean atmospheric $NO_3^-$"